# Multi-Step Reasoning for Embodied Question Answering via Tool Augmentation

**Question**

Which size is larger, the brown sofa in the living room or the red sofa in the study room?

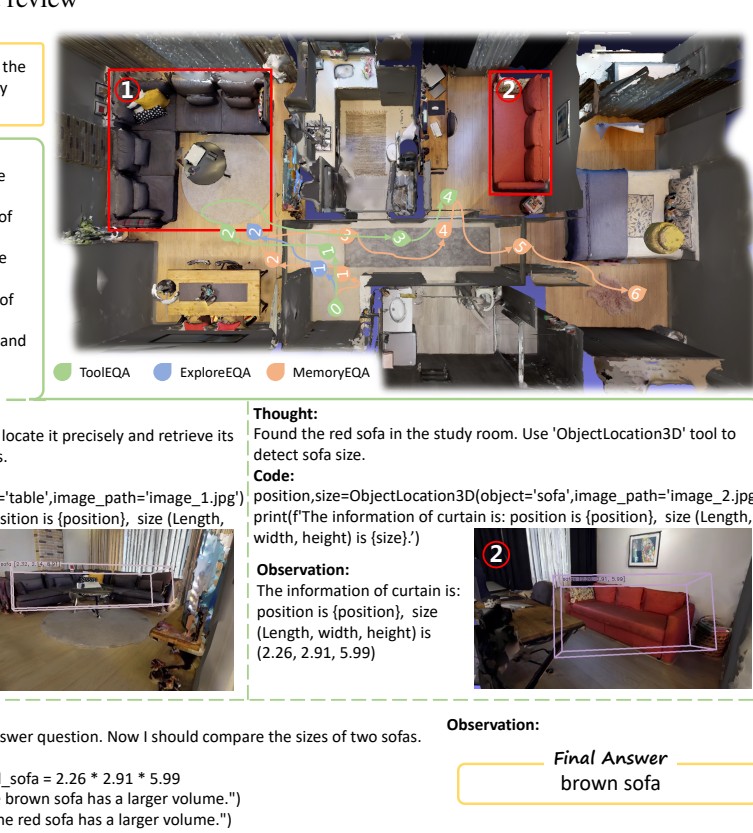

**ToolEQA**

**Plan:**
1. Go to the living room and find the brown sofa.
2. Check the size in the living room of the brown sofa.
3. Go to the study room and find the red sofa.
4. Check the size in the study room of the red sofa.
5. Compare the sizes of brown sofa and red sofa.
6. Provide the final answer.

**Thought:**
Found the sofa in the living room. I will locate it precisely and retrieve its bounding box to confirm its dimensions.
**Code:**
position,size=ObjectLocation3D(object='table',image_path='image_1.jpg')
print(f'The information of curtain is: position is {position}, size (Length, width, height) is {size}.')
**Observation:**
The information of curtain is: position is {position}, size (Length, width, height) is (2.32, 2.14, 8.91)

**Thought:**
Found the red sofa in the study room. Use 'ObjectLocation3D' tool to detect sofa size.
**Code:**
position,size=ObjectLocation3D(object='sofa',image_path='image_2.jpg')
print(f'The information of curtain is: position is {position}, size (Length, width, height) is {size}.')
**Observation:**
The information of curtain is: position is {position}, size (Length, width, height) is (2.26, 2.91, 5.99)

**Thought:**
I already know all information for answer question. Now I should compare the sizes of two sofas.
**Code:**
```
brown_sofa = 2.32 * 2.14 * 8.91; red_sofa = 2.26 * 2.91 * 5.99
if brown_sofa > red_sofa : print("the brown sofa has a larger volume.")
elif brown_sofa < red_sofa : print("the red sofa has a larger volume.")
final_answer('brown sofa')
```

**Observation:**

**Final Answer**
brown sofa

Figure 1: Overview of the proposed ToolEQA for Embodied Question Answering (EQA). ToolEQA enables to decompose questions into structured plans, reasoning to select tools, and invoke tools to explore and answer. ToolEQA achieves highest accuracy with fewer reasoning steps.

## Abstract

Embodied Question Answering (EQA) requires agents to explore 3D environments to obtain observations and answer questions related to the scene. Existing methods leverage VLMs to directly explore the environment and answer questions without explicit thinking or planning, which limits their reasoning ability and results in excessive or inefficient exploration as well as ineffective responses. In this paper, we introduce **ToolEQA**, an agent that integrates external tools with multi-step reasoning, where external tools can provide more useful information for completing the task, helping the model derive better exploration directions in the next step of reasoning and thus obtaining additional effective information. This enables ToolEQA to generate more accurate responses with a shorter exploration distance. To enhance the model's ability for tool-usage and multi-step reasoning, we further design a novel EQA data generation pipeline that automatically constructs large-scale EQA tasks with reasoning trajectories and corresponding answers. Based on the pipeline, we collect the EQA-RT dataset that contains about 18K tasks, divided into a training set EQA-RT-Train, and two test sets EQA-RT-Seen (scenes overlapping with the training set) and EQA-RT-Unseen (novel scenes). Experiments on EQA-RT-Seen and EQA-RT-Unseen show that ToolEQA improves the success rate by 9.2∼20.2% over state-of-the-art baselines, while outperforming the

zero-shot ToolEQA by 10% in success rate. In addition, ToolEQA also achieves state-of-the-art performance on the HM-EQA, OpenEQA, and EXPRESS-Bench datasets, demonstrating its generality.

# 1 INTRODUCTION

Embodied Question Answering (EQA), a challenging task in computer vision and robotics, requires agents to navigate in a 3D environment, actively gather visual information through exploration, and answer questions about the scene (Das et al., 2018a). Existing methods (Ziliotto et al., 2025; Ren et al., 2024a; Zhai et al., 2025; Cheng et al., 2024; Jiang et al., 2025) leverage VLMs to understand environment for guiding exploration and answering questions, but they generally lack explicit intermediate reasoning and planning. For example, as shown in Figure 1, (1) The agent often answers the question before fully identifying all relevant objects, resulting in incorrect final answers due to insufficient information gathering capabilities. (2) The agent makes suboptimal route plans, prolonging the exploration process and reduce efficiency due to limited reasoning abilities. This motivates us to leverage tools to enhance the information-gathering capabilities of the agent, and use multi-step explicit reasoning to improve its reasoning ability during the exploration process, enabling it to complete EQA tasks with more efficient exploration distances.

In this paper, we propose ToolEQA, an agent that leverages tool augmentation to perform multi-step reasoning for EQA tasks. ToolEQA reasons over both current observations and historical information, selects appropriate tools to invoke, and integrates the additional information they provide (e.g., 3D bounding boxes) into the reasoning process. To ground reasoning in the environment, we abstract the action space into tool sets and execute them as actions. The agent iteratively reasons and applies tools, acquiring new observations until the final answer is derived. By effectively integrating collected information and identifying shorter exploration paths, ToolEQA improves both exploration efficiency and accuracy in solving EQA tasks.

To enhance the reasoning capability of the ToolEQA agent, we introduce a novel EQA data generation pipeline that automatically generates large-scale EQA tasks with reasoning trajectories via three steps: EQA task generation, reasoning trajectory generation, and validation. Specifically, we first employ a 3D detection model to identify all objects in the current scene and extract their attributes, such as size and spatial coordinates. Based on this object-level information, we then leverage GPT-4o (OpenAI, 2024a) to automatically generate diverse questions and their corresponding answers. Subsequently, to generate optimal reasoning trajectories, we extract all relevant objects mentioned in the question and determine the shortest path by combining their positions with an A-star algorithm. On top of this path, we incorporate reasoning steps and tool usage into the path by employing GPT-4o to generate complete trajectories. To ensure the correctness of questions, we design question-type-specific prompt templates that guide the generation process, thereby ensuring both path optimality and consistency in task solving. Finally, to preserve data quality, the generated EQA tasks and trajectories are passed through an EQA task verifier and trajectory verifier to discard low-quality data and rectify incorrect trajectories.

With the data generation pipeline, we construct EQA-RT, a dataset of 18K EQA question–answer pairs with reasoning trajectories. We further split it into a training set (EQA-RT-Train) and two test sets, where two test sets contain EQA-RT-Seen (in-domain scenes overlapping with the training set) and EQA-RT-Unseen (out-of-domain scenes for evaluating generalization). We train the proposed ToolEQA agent on EQA-RT-Train using supervised fine-tuning. We comprehensively evaluate the tuned ToolEQA agent and the zero-shot ToolEQA agent on HM-EQA (Ren et al., 2024a), Open-EQA (Majumdar et al., 2024), ExpressBench (Jiang et al., 2025), EQA-RT-Seen and EQA-RT-Unseen. The ToolEQA agent consistently achieves improvements on untrained VLMs and outperforms them by 11%. This indicates that our method enables agents to have powerful capability for practical EQA tasks with complex and diverse trajectories. In summary, our contributions are three-fold.

- We propose the ToolEQA agent which performs multi-step reasoning for environment exploration and question answering, achieving improved effectiveness and efficiency in solving EQA tasks.

- We introduce an EQA data generation pipeline that automatically generates large-scale EQA tasks with reasoning trajectories.

- We introduce EQA-RT, a dataset containing 18K question–answer pairs for EQA, covering diverse and complex question types with high-quality reasoning trajectories.

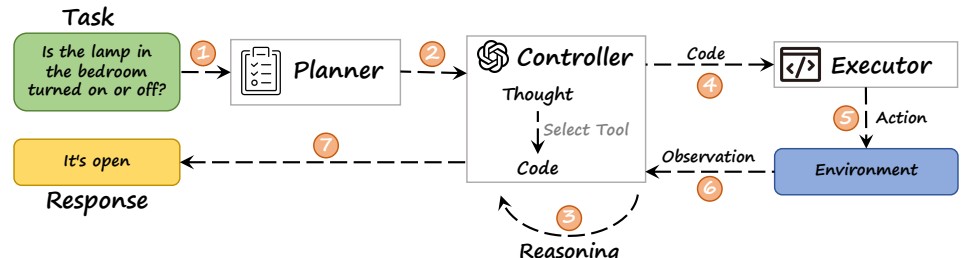

Figure 2: Overview of the ToolEQA agent workflow.

## 2 RELATED WORKS

### 2.1 EMBODIED QUESTION ANSWERING

Embodied question answering (Das et al., 2018a; Gordon et al., 2018; Yu et al., 2019; Cangea et al., 2019; Das et al., 2018b) has become a challenging paradigm for testing a robot's ability to autonomously plan tasks and establish semantic understanding of the environment in order to correctly answer natural language questions. Yu et al. (2019) constructed a multi-target question answering dataset in a virtual environment and introduced a multi-target EQA method. Ren et al. (2024a) first applied VLMs to EQA and built the HM-EQA dataset with more open-ended questions for realistic and diverse evaluation. Subsequent studies extended VLMs for EQA. Majumdar et al. (2024) used video memory for implicit questions with long-context VLMs. Saxena et al. (2024) embedded a planner into compact scene representations to bridge semantic memory and planning. Jiang et al. (2025) added exploration-trajectory annotations to EQA datasets and incorporated exploration into evaluation metrics. However, those methods demonstrate limited reasoning capacity, as they lack explicit thinking and planning, which often leads to redundant or inefficient exploration. To solve this problem, we introduce a multi-step reasoning process for solving EQA task via tool augmentation, enhancing effectiveness and efficiency.

### 2.2 MULTI-STEP REASONING

Multi-step reasoning can significantly enhance a model's ability to solve complex tasks while improving interpretability. In recent years, research on multi-step reasoning in large language models (LLMs) (Ranaldi et al., 2024; OpenAI, 2024b; Chen et al., 2024; Yao et al., 2025) and multi-modal systems has made notable progress. Li et al. (2024) proposed the VoCoT framework, which integrates vision-guided and object-centric chain-of-thought reasoning to improve the reasoning performance of large-scale multi-modal models on complex tasks. The ReAct (Yao et al., 2023) framework is a multi-step reasoning paradigm that decomposes tasks through iterative Reason–Act–Observe cycles, which greatly benefits to solving challenging problems. In the domain of Embodied Question Answering (EQA), Fine-EQA (Jiang et al., 2025) introduced a new benchmark that emphasizes dynamic exploration and multi-step reasoning in 3D environments, aiming to improve both exploration efficiency and evaluation metrics. However, the exploration of multi-step reasoning in embodied question answering remains at a relatively early stage. We propose ToolEQA, which integrates explicit multi-step reasoning into the multi-step exploration process in embodied scenarios. This step-by-step thinking strategy not only shortens exploration paths but also enhances the accuracy of question answering.

### 2.3 TOOL USAGE AGENT

Recent work have equipped VLMs with tool-usage capabilities. Frameworks such as ReAct (Yao et al., 2023) and Toolformer (Schick et al., 2023) demonstrated the effectiveness of coupling reasoning traces with tool execution, while embodied agents like SayCan (Ahn et al., 2022) showed how language-guided tool usage can translate high-level instructions into low-level actions. T3-Agent (Gao et al., 2024) leveraged automatically generated multi-modal tool-usage data and fine-tuned vision-language models (VLMs) as controllers to enable strong tool-based reasoning. Li et al. (2025) proposed the MeCo framework, which captures the model's "cognitive signals" to assess its capability boundaries and thereby decide whether to invoke external tools. These works enhance multi-modal reasoning by calling predefined tools to acquire additional information. However, such methods perform reasoning only within static cyberspace. In contrast, we define the physical environment itself as a tool, thereby situating reasoning steps within embodied interactions and enabling more autonomous embodied agents.

## 3 TOOLEQA AGENT

To enable the agent to reason and act in complex environments, we propose a ToolEQA agent that integrates tool-usage strategies for the reasoning process. ToolEQA conducts step-by-step reasoning on past observations and, at each step, generates corresponding thoughts and code to execute tools. Code offers greater flexibility than formats such as JSON for handling diverse inputs and outputs. As shown in Figure 2, ToolEQA comprises three components: a planner for generating overall task plan $p$, a controller for generating thought $t$ and code $c$, and an executor for executing code in environment. Given a query $Q$ and a scene $S$, the $i$-step of the agent is formulated as

$$t_i^*, c_i^* = \arg\max P(t_i, c_i|Q, S, h_i, p), \tag{1}$$

where $t_i^*$ and $c_i^*$ are generated thought and code for the $i$-th step, and $h_i = \{t_1, c_1, o_1, ..., t_{i-1}, c_{i-1}, o_{i-1}\}$ is the history (thought, code, and observation of previous steps).

**Planner.** Given an EQA task, the planner, modeled as an LLM, takes the query as input, interprets the task objectives, and outputs an overall plan that decomposes the task into sub-goals. The structured sub-goals are provided to the controller to prevent blind exploration, enhancing the efficiency and accuracy of task execution.

**Executor.** We deploy real-executable tools for the agent. Our tools include `GoNextPoint`, `ObjectLocation2D`, `ObjectLocation3D`, `ObjectCrop`, `VisualQA`, `FinalAnswer`, the details of tools see Appendix A.4. With the generated code, the executor calls executable tools in the environment to obtain new observations for further exploration, thus solving the EQA task.

**Controller.** The controller performs dynamic reasoning to decide which tool to use, based on the question, previous observations, and the guidance of plans, and invokes the executor to gather new observations for further reasoning until the answer is derived. The reasoning process can be divided into three situations.

- **The collected information is insufficient and the current scene lacks required objects.** ToolEQA infers missing objects from previous observations and the query, and estimates their likely positions. Then, ToolEQA combines these estimates with its current location to decide a walking direction, and uses the '`GoNextPoint`' (for example, '`GoNextPoint("turn left")`') to gather the needed information.

- **The collected information is insufficient and the current scene contains involved objects.** ToolEQA reasons over the question and invokes suitable tools to obtain relevant information. For example, as shown in Figure 1, ToolEQA utilizes '`ObjectLocation3D`' to extract the size of objects, and then continues exploration following the above step.

- **The collected information is sufficient.** ToolEQA needs to reason over the question, use appropriate tools on the current image, and integrate information for the final answer. For example, it processes the detected object, then writes 'Python' codes for comparing sizes, and employs '`FinalAnswer`' to produce the ultimate output.

## 4 EQA DATA GENERATION PIPELINE

### 4.1 FORMULATION

**Data Format.** We format the EQA tool-usage data as $\{S, pos, Q, p, T, C, O, A\}$, where $S$ denotes the scene, $pos$ denotes the initial position of agents, $Q$ denotes the question, $p$ denotes the overall plan for solving the task, $T$ denotes the generated thought, $C$ denotes the generated code, $O$ denotes observation (outputs of using tools), and $A$ means the ground truth answer. Considering that solving one real-world EQA task may require multiple steps involving multiple tools, $T$, $C$, and $O$ can be represented by the integration of thought, code, and observation in multiple steps, and the data format is reformulated as $\{S, pos, Q, p, \{t_1, \cdots, t_n\}, \{c_1, \cdots, c_n\}, \{o_1, \cdots, o_n\}, A\}$, where $t_i, c_i$, and $o_i$ indicate the thought, code, and observation in the $i$-th step respectively, and there are $n$ steps in total. The thought, code, and observation are composed of a trajectory $\{t_1, c_1, o_1, \cdots, t_n, c_n, o_n\}$ of $n$ steps to solve the task.

**Scene Source.** HM3D (Ramakrishnan et al., 2021) is a comprehensive dataset comprising 3D reconstructions of 1,000 large-scale buildings collected from diverse real-world locations. We select 713 high-quality scenes from HM3D as our data source, sample object images from them, and generate questions and answers.

The proposed data generation pipeline is shown in Figure 3, including three steps: EQA task generation, reasoning trajectory generation, and data verification.

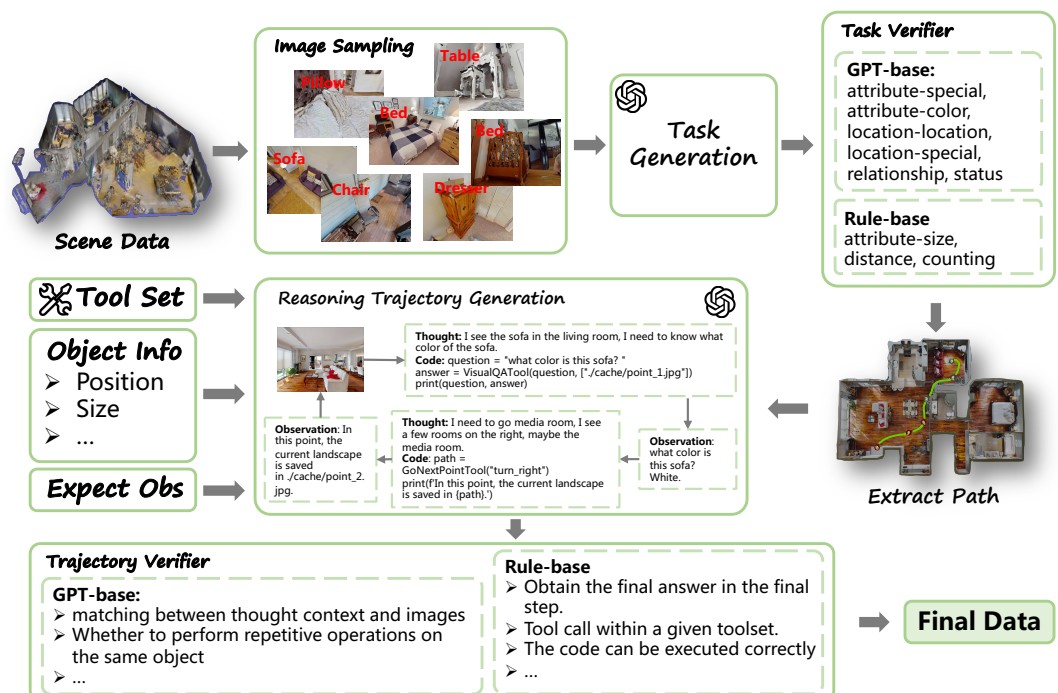

Figure 3: EQA Data Generation Pipeline.

## 4.2 EQA TASK GENERATION

Our goal is to generate a large set of diverse, practical, and complex EQA tasks. We first apply a 3D detection model to obtain each object's bounding box, position, and category, and sample the object image from detected objects. The object attributes and corresponding visual information are then fed into GPT-4o along with example question-answer pairs designed from brainstorming to simulate natural home conversations. Guided by the prompt, GPT-4o generates questions and answers across six types: relationship, status, distance, location, counting, and attribute, where location is divided into two subcategories 'location-location' and 'location-special', and attribute is divided into three subcategories 'color', 'special', and 'size'. The answers are open-ended or multiple-choice, enabling the evaluating different capabilities of agents.

## 4.3 REASONING TRAJECTORY GENERATION

Given an EQA task, we construct an exploration trajectory that records reasoning steps, tool selections, and observations. The trajectory is constrained to follow the shortest path and ensure consistency between reasoning and tool usage. We extract objects mentioned in the question using their locations and the agent's position, and compute the shortest path using the A* algorithm, generating intermediate waypoints and navigation directions.

Based on these trajectories, GPT-4o enriches each step with reasoning and tool selections. Steps are categorized as key, where the target object is found, and non-key, where it is not. For non-key steps, GPT-4o receives the current image and exploration direction to generate reasoning. For the key steps, we select possible tools required to solve the task from the toolset, and then prompt GPT-4o to output which specific tool should be invoked under the current observation and the corresponding rationale. To ensure consistency and rationality, we design question-type–specific prompts containing task-specific considerations, reasoning strategies, tool-usage guidelines, and examples, allowing GPT-4o to produce thought and code across different question types.

## 4.4 DATA VERIFICATION

To preserve the quality of generated data, we design an EQA task verifier and a trajectory verifier to filter out low-quality data. Using LLMs to verify generated tasks and trajectories has proven effective (Gao et al., 2024; Liu et al., 2024). Inspired by this, we use LLMs to verify generated tasks and trajectories.

**EQA Task Verifier.** Since object descriptions in generated questions or options may not always match the scene, we use two complementary strategies: confidence-based matching and LLM-based structured scoring to evaluate quality and filter out low-quality samples. For confidence, we first extract object descriptions from the question and options, locate the corresponding objects in the scene to obtain their images, and then use Grounded-SAM (Ren et al., 2024b) to compute a score reflecting how well each image matches its description. For LLM-based scoring, we feed the question and object images into GPT-4o, which outputs a similarity score. We set thresholds for the two strategies respectively, and samples below thresholds on either score are filtered out.

**Trajectory Verifier.** To verify the rationality of tool usage and reasoning in the generated trajectories, we adopt two strategies: rule-based and LLM-based validation. For the rule-based validation, we design several checks: (1) the key tools should exist and be invoked at the correct step (*e.g.*, `GoNextPoint` should be called at every step before reaching the target); (2) the invoked tool should belong to the predefined tool set, and its parameters should be passed correctly. For the LLM-based validation, we prompt GPT-4o to consider the following factors: (1) the predicted answer should be semantically consistent with the ground-truth answer; (2) the reasoning in non-key steps should avoid hallucinations; (3) the final answer should be reasonably derived from the reasoning and observations. We set a threshold for three checks, and samples below the threshold are filtered out.

## 4.5 EQA-RT

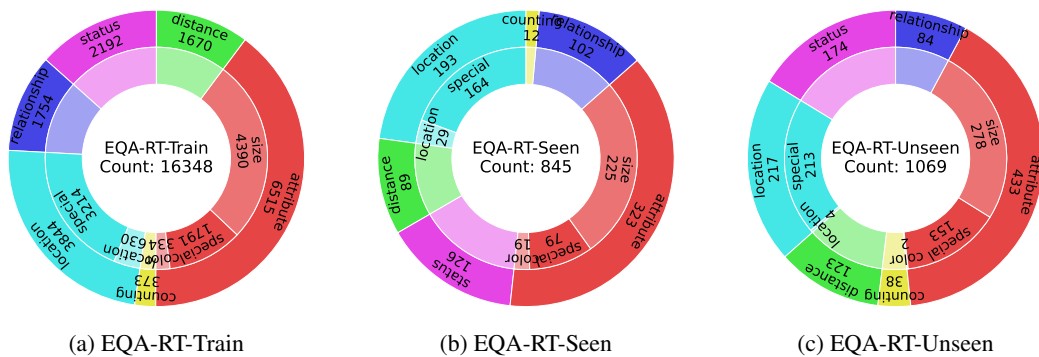

(a) EQA-RT-Train      (b) EQA-RT-Seen      (c) EQA-RT-Unseen

Figure 4: Data statistics of the training set (EQA-RT-Train) and two test sets (EQA-RT-Seen and EQA-RT-Unseen). The scenes in EQA-RT-Seen have the overlap with EQA-RT-Train, while the scenes in EQA-RT-Unseen are not present in the training set.

By utilizing the developed EQA task generation pipeline, we construct EQA-RT, encompassing about 18K EQA tasks. We further split it into a training set (EQA-RT-Train), a seen test set (EQA-RT-Seen) and a unseen test set (EQA-RT-Unseen), where the test set contains both in-domain scenes overlapping with the training set and out-of-domain scenes for evaluating generalization. As shown in Figure 4, we show the question types of the generated EQA tasks in EQA-RT-train, EQA-RT-Seen and EQA-RT-Unseen. More statistical data can be found in Appendix A.1.

## 4.6 TRAINING

Given a data point $\{S, pos, Q, p\{t_1, ..., t_n\}, \{c_1, ..., c_n\}, \{o_1, ..., o_n\}, A\}$, we train the VLM controller using the cross-entropy loss,

$$\min \mathbb{E}_{(Q,S,pos,T,C,O,A)\sim\mathbb{D}} \left[ -\sum_{i=1}^{n} P(t_i, c_i | Q, S, pos, h_i) \right], \tag{2}$$

where $\mathbb{D}$ is the EQA-RT-Train dataset and we sum the loss values of the n steps in the trajectory. Note that, in training VLMs, we do not fit the final answer $A$, as we encourage the controller to leverage tools in solving given tasks, instead of directly producing an answer based on biases in VLMs. The average length of exploration and reasoning trajectories reaches 12.69 steps (as shown in the statistics in Appendix Table 6). A longer number of steps results in an extended trajectory history $h$, which in turn enlarges the model input and ultimately causes substantial time and memory consumption during training. To address this issue, we propose a trajectory sampling strategy. Specifically, we retain the key steps and randomly sample an equal number of non-key steps to

Table 1: Baseline evaluation on EQA-RT-Seen.

| Setting | Model | recall ↑ | | | $e_{path}$ ↑ | | | succ. (%) ↑ |
|---|---|---|---|---|---|---|---|---|
| | | @5 | @10 | @15 | @5 | @10 | @15 | |
| Multi Choices | Explore-EQA | 0.06 | 0.11 | 0.14 | 0.04 | 0.07 | 0.09 | 44.7 |
| | Memory-EQA | 0.06 | 0.12 | 0.13 | 0.04 | 0.07 | 0.11 | 48.2 |
| | ToolEQA (gpt-4o) | 0.06 | 0.14 | 0.19 | **0.08** | 0.2 | 0.27 | 55.37 |
| | ToolEQA (qwen2.5vl) | 0.04 | 0.09 | 0.11 | 0.06 | 0.13 | 0.17 | 53.1 |
| | ToolEQA (qwen2.5vl ft) | **0.06** | **0.15** | **0.21** | 0.07 | **0.23** | **0.3** | **57.31** |
| Open Vocabulary | Explore-EQA | 0.04 | 0.10 | 0.13 | 0.04 | 0.06 | 0.09 | 30.6 |
| | Memory-EQA | 0.05 | 0.10 | 0.13 | 0.04 | 0.09 | 0.11 | 35.1 |
| | ToolEQA (gpt-4o) | 0.06 | 0.15 | **0.21** | 0.07 | 0.2 | 0.27 | 49.2 |
| | ToolEQA (qwen2.5vl) | 0.05 | 0.12 | 0.16 | 0.03 | 0.10 | 0.14 | 44.9 |
| | ToolEQA (qwen2.5vl ft) | **0.06** | **0.15** | 0.20 | **0.08** | **0.22** | **0.3** | **53.6** |

Table 2: Baseline evaluation on EQA-RT-Unseen.

| Setting | Model | recall ↑ | | | $e_{path}$ ↑ | | | succ. (%) ↑ |
|---|---|---|---|---|---|---|---|---|
| | | @5 | @10 | @15 | @5 | @10 | @15 | |
| Multi Choices | Explore-EQA | 0.06 | 0.12 | 0.15 | 0.05 | 0.08 | 0.10 | 47.0 |
| | Memory-EQA | 0.06 | 0.13 | 0.14 | 0.06 | 0.09 | 0.11 | 48.9 |
| | ToolEQA (gpt-4o) | 0.07 | **0.16** | **0.21** | 0.08 | 0.2 | 0.28 | 57.9 |
| | ToolEQA (qwen2.5vl) | 0.04 | 0.11 | 0.13 | 0.07 | 0.16 | 0.21 | 55.3 |
| | ToolEQA (qwen2.5vl ft) | **0.07** | 0.14 | 0.19 | **0.08** | **0.24** | **0.3** | **59.5** |
| Open Vocabulary | Explore-EQA | 0.05 | 0.09 | 0.15 | 0.05 | 0.10 | 0.13 | 31.4 |
| | Memory-EQA | 0.05 | 0.09 | 0.15 | 0.06 | 0.15 | 0.18 | 35.9 |
| | ToolEQA (gpt-4o) | 0.06 | 0.16 | 0.21 | 0.06 | 0.21 | 0.27 | 49.3 |
| | ToolEQA (qwen2.5vl) | 0.06 | 0.13 | 0.17 | 0.05 | 0.16 | 0.2 | 45.1 |
| | ToolEQA (qwen2.5vl ft) | **0.08** | **0.17** | **0.24** | **0.09** | **0.24** | **0.32** | **56.1** |

reduce resource overhead. This design is motivated by the fact that non-key steps dominate the exploration process and are highly redundant, and they mostly consist of repeated direction predictions and frequent use of the `GoNextPoint` tool. In contrast, key steps involve diverse tool usage and reasoning changes. After training, ToolEQA agent can present powerful ability of reasoning and tool-usage, further enhance the effectiveness and efficiency of solving EQA tasks.

## 5 EXPERIMENTS

### 5.1 SETTING

We tested ToolEQA on the EQA-RT and HM-EQA (Ren et al., 2024a) datasets and compared it with existing open-source methods, Explore-EQA (Ren et al., 2024a) and Memory-EQA (Zhai et al., 2025). We also examined the impact of different models (GPT-4o (OpenAI, 2024a), Qwen2.5-VL-7B (Wang et al., 2024), and fine-tuned Qwen2.5-VL-7B (Wang et al., 2024)) as controllers on performance. In addition, we conducted a qualitative analysis of ToolEQA, investigating how reasoning and tool invocation affect the efficiency and success rate of completing EQA tasks.

**Training** We trained the controller using the EQA-RT training set. During the training of the VLM-based controller, we froze the vision encoder and the visual token compressor, and fine-tuned the language model with LoRA (Hu et al., 2022). We adopted the AdamW optimizer with a cosine annealing scheduler, using a learning rate of 1e-6 and a batch size of 1. We used 4 Nvidia Tesla H100 GPUs to train for 2 days.

**Metrics** We use three metrics for evaluating ToolEQA and existing EQA methods. The success rate is divided into two parts, for multi-choices tasks, we calculate average accuracy between the output of model and ground truth answer; for open vocabulary tasks, we prompt LLM to obtain the semantic similarity between the output of model and ground truth answer. $recall@D$ is used to evaluate whether objects related to the problem were found during the exploration process. $e_{path}@D$ is an indicator that combines success rate, recall, and exploration path length. The details of the metrics can be found in Appendix A.3.

### 5.2 MAIN RESULTS

As shown in Table 1 and Table 2, we report the performance of different methods on EQA-RT-Seen and EQA-RT-Unseen. Our ToolEQA consistently outperforms reasoning-inefficient methods Explore-EQA (Ren et al., 2024a) and Memory-EQA (Zhai et al., 2025) across all metrics, demonstrating its effectiveness in tackling complex tasks. The comparison between agents equipped with fine-tuned and non-fine-tuned VLMs further validates the effectiveness of our data generation pipeline. The success rate of fine-tuned Qwen2.5VL-7B compared to the original Qwen2.5VL-7B on EQA-RT-Unseen improved from 45.1 to 56.1, the recall rate increased from 0.17 to 0.24, and $e_{path}$ improved from 0.2 to 0.32. Compared with the non-fine-tuned Qwen2.5-VL-7B, ToolEQA with GPT-4o achieves better performance, indicating that the controller's capability directly determines the performance of ToolEQA. However, the fine-tuned Qwen2.5-VL-7B surpasses GPT-4o in $e_{path}$ and success rate, while achieving comparable recall. This indicates that our training has enabled the VLM to learn how to think and solve problems more effectively in indoor scenarios.

Table 3: Comparison between the original model and the finetuned model in terms of the number of key steps, thought length, correct tool usage rate, and success rate on the EQA-RT-Unseen dataset.

| Model | Step | Thought | Tool (%) | Succ. |
|---|---|---|---|---|
| ToolEQA (0-shot) | 1.24 | 90.26 | 58 | 45.1 |
| ToolEQA (ft) | **1.98** | **116.15** | **69** | **56.1** |

Table 4: Performance comparison on EXPRESS-Bench.

| Model | Succ. ↑ | Succ.* ↑ | $E_{path}$ ↑ | $d_T$ ↓ |
|---|---|---|---|---|
| Fine-EQA | 40.55 | 63.95 | 16.22 | 6.43 |
| ToolEQA(0-shot) | 40.65 | 65.01 | 22.33 | 6.31 |
| ToolEQA(ft) | **42.21** | **65.77** | **25.82** | **5.25** |

Table 5: EQA-Agent performance on existed benchmarks. † represents that the metric comes from the our implementation.

| Model | HM-EQA | | OpenEQA | |
|---|---|---|---|---|
| | succ.(%) | L(m) | succ.(%) | L(m) |
| Explore-EQA | 51.5 | 38.87 | 28.3† | 25.45 |
| Efficient-EQA | 54.3 | 30.16 | - | - |
| Memory-EQA | 63.4 | 33.54 | 34.6† | 11.32 |
| Fine-EQA | 53.3† | 34.75† | 29.4† | 13.77 |
| Graph-EQA | **63.5** | - | 30.1† | 11.96 |
| ToolEQA(0-shot) | 61.0 | 18.98 | 33.1 | 8.38 |
| ToolEQA(ft) | 62.3 | **18.26** | **35.5** | **6.96** |

We typically consider that the length of thoughts is positively correlated with reasoning ability (Jin et al., 2024). Therefore, we evaluate the impact of fine-tuning the VLM on EQA-RT-Unseen with respect to thought length and the accuracy of tool calls (i.e., the proportion of calls that successfully acquire the information required to answer the question). As shown in Table 3, after fine-tuning, thought length increases from 90.26 to 116.15, tool call accuracy improves from 58 to 69, and the success rate rises from 45.1 to 56.1. This indicates that reasoning ability is crucial for accomplishing the EQA task.

As shown in Table 5, we compare our method with Fine-EQA (Jiang et al., 2025) on EXPRESS-Bench (Jiang et al., 2025), and with Explore-EQA (Ren et al., 2024a), Efficient-EQA (Cheng et al., 2024), Memory-EQA (Zhai et al., 2025), and Graph-EQA (Saxena et al., 2024) on HM-EQA (Ren et al., 2024a) and OpenEQA (Majumdar et al., 2024). Our method achieves comparable success rates with shorter exploration distances across all these datasets.

## 5.3 QUALITATIVE RESULTS

As shown in Figure 5, the decision-making process integrates spatial layout, functional space, and environmental cues to guide navigation toward the target object (towels). Each step is supported by clear reasoning, such as moving forward to approach potential bathroom space, turning right to explore a promising corridor, or turning left after excluding non-target rooms. This information-driven and reasoning-based decision paradigm ensures that the generated exploration trajectory maintains a high degree of proximity to the ground-truth trajectory, effectively validating the rationality and effectiveness of the decision-making framework in target-oriented spatial exploration tasks.

Figure 6 highlights the clear advantage of tool-driven reasoning over direct Visual-Language Model (VLM) inference. Without tools, VLM often fails to localize objects precisely or distinguish fine-grained attributes such as size. In contrast, by integrating these specialized tools, our method obtained critical, fine-grained information (precise object localization, clutter-free cropping, and accurate size identification) that cannot be reliably captured by direct VLM inference on unprocessed images. This structured tool usage ensured the final comparison concluding that "The chair has

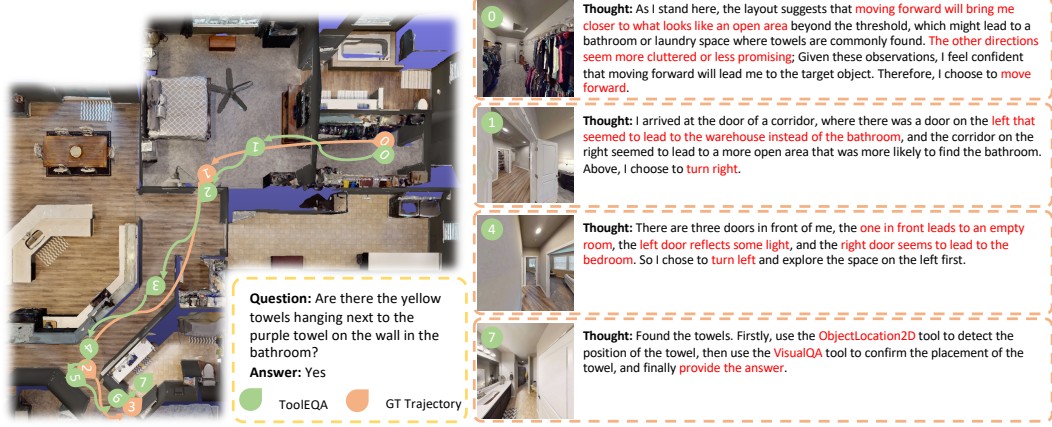

Figure 5: Illustration of how explicit reasoning guides efficient exploration, enabling ToolEQA to answer questions faster and more accurately.

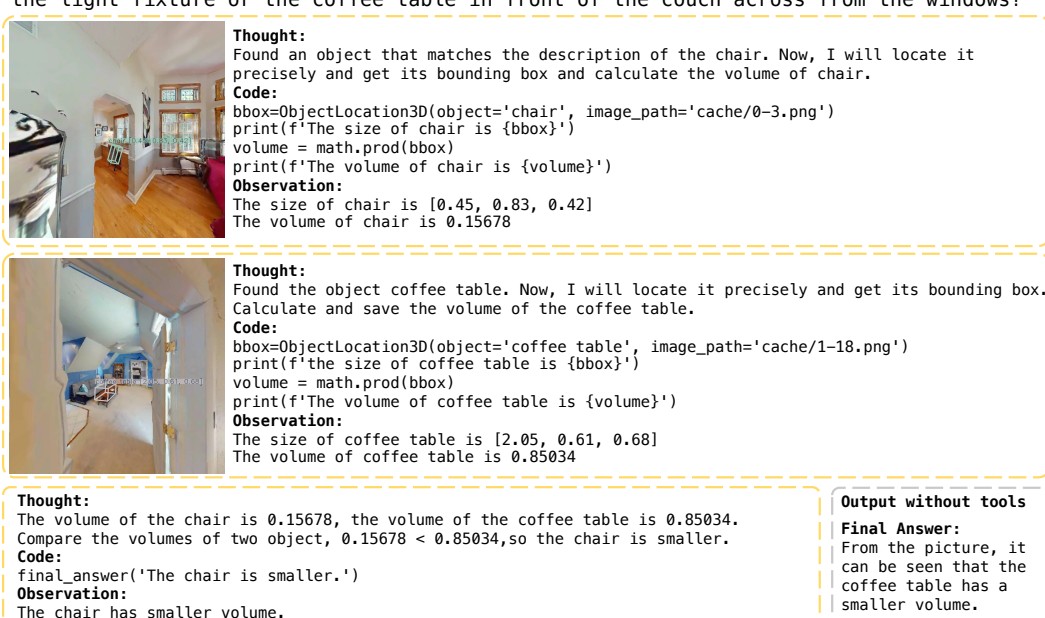

Figure 6: Demonstration that the visual tools outperforms direct VLM inference without tools by accurately localizing, and comparing object volume.

smaller volume" was grounded in objective data, ultimately achieving a more accuracy response than would be possible with VLM alone.

## 6 CONCLUSION

In this work, we introduced **ToolEQA**, an embodied question answering agent that integrates explicit multi-step reasoning with tool usage to guide exploration and information acquisition. By coupling dynamic reasoning with executable tools, ToolEQA enables more efficient exploration paths and more reliable utilization of gathered observations. To support training, we proposed a scalable EQA data generation pipeline and constructed **EQA-RT**, a dataset of 18K automatically generated tasks with validated reasoning trajectories. Comprehensive experiments on HM-EQA, OpenEQA, ExpressBench and EQA-RT demonstrate that ToolEQA achieves significant improvements in both accuracy and efficiency over prior methods. These results highlight the importance of explicit multi-step reasoning and tool-usage in EQA agents, and suggest promising directions for developing more generalizable and interpretable frameworks for complex embodied AI tasks.

## 7  ETHICS STATEMENT

This research adheres to the ICLR Code of Ethics. We confirm that all aspects of the study were conducted with the highest ethical standards. The work does not involve human subjects, and no personally identifiable or sensitive data were used. All datasets (HM3D, HM-EQA, OpenEQA, ExpressBench and EQA-RT) utilized were publicly available and comply with privacy regulations. We have taken steps to ensure that the models and algorithms developed are fair and free from bias. No conflicts of interest or external sponsorship influenced this work. Additionally, the findings are presented honestly, with all methodologies thoroughly validated for accuracy and reproducibility. We believe this research aligns with the ethical guidelines set forth by ICLR and contributes to the integrity and transparency of the scientific community.

## 8  REPRODUCIBILITY STATEMENT

To ensure the reproducibility of our results, we provide an anonymous code repository `https://anonymous.4open.science/r/ReactEQA-DDE3` that includes the complete implementation of data generation, data validation, and the ToolEQA framework. The repository also contains a comprehensive `README.md` file that details the installation requirements, dataset preparation, and step-by-step execution instructions. By following these guidelines, all experiments and results reported in this paper can be reproduced.

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

# A APPENDIX

## A.1 DATA STATISTIC

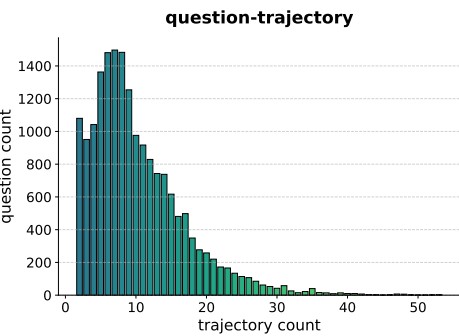
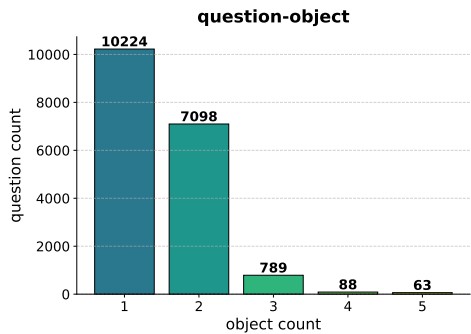

(a) Statistics on the number of questions with different steps counts.

(b) Statistics on the number of questions with different object counts.

Figure 7: All data statistic on Thought Step and Object.

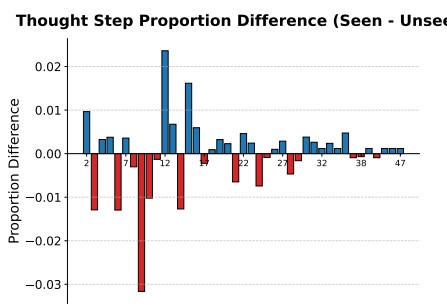
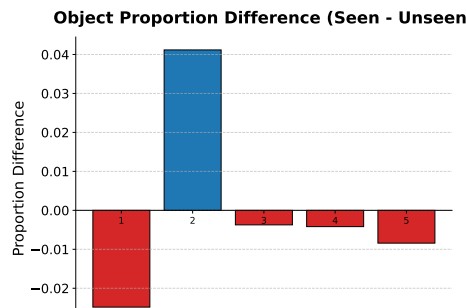

(a) The difference in the ratio of the average thought steps between the EQA-RT-Seen and the EQA-RT-Unseen.

(b) The difference in the ratio of the average number of objects between the EQA-RT-Seen and the EQA-RT-Unseen.

Figure 8: Comparison of EQA-RT-Seen and EQA-RT-Unseen on Thought Step and Object.

Table 6: Statistic about average exploration steps, the average number of tools used per task, and the average exploration length.

|  | Step | Tool | Length (m) |
| --- | --- | --- | --- |
| EQA-RT-Train | 12.74 | 12.39 | 13.13 |
| EQA-RT-Seen | 12.56 | 12.20 | 12.71 |
| EQA-RT-Unseen | 12.13 | 11.76 | 12.38 |
| EQA-RT | 12.69 | 12.35 | 13.07 |

As shown in Figure 7, the dataset exhibits a pronounced long-tail distribution in both the average exploration steps and the number of related objects per question. Most questions require around ten exploration steps; among them, 10,224 involve a single target, 7,098 involve two targets, and 940 involve three or more objects. Figure 8(a) further compares EQA-RT-Seen and EQA-RT-Unseen in terms of the average number of objects per question, revealing that EQA-RT-Seen involves more objects. And Figure 8(b) compares EQA-RT-Seen and EQA-RT-Unseen in terms of the average thought steps per question, revealing that EQA-RT-Unseen involves more thought steps.

In addition, as shown in Table 6, we report the exact values of the average exploration steps, the average number of tools used per task, and the average exploration length across different sets. By

comparing the statistics of EQA-RT-Seen and EQA-RT-Unseen, it can also be inferred that the tasks in EQA-RT-Seen are more challenging.

## A.2 RESULT ANALYSIS

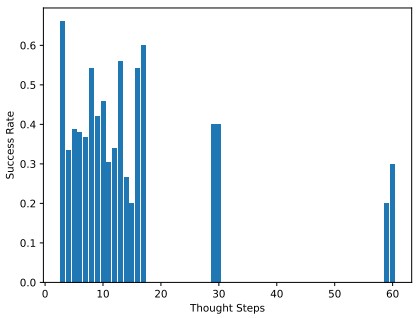

(a) Success rate with different thought steps on EQA-RT-Seen.

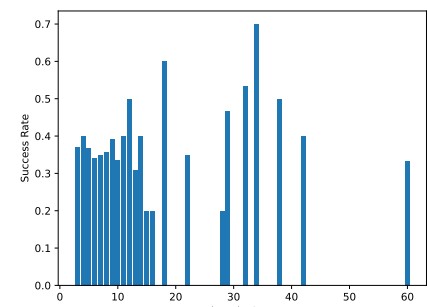

(b) Success rate with different thought steps on EQA-RT-Unseen.

Figure 9: The relationship between thought steps and success rate.

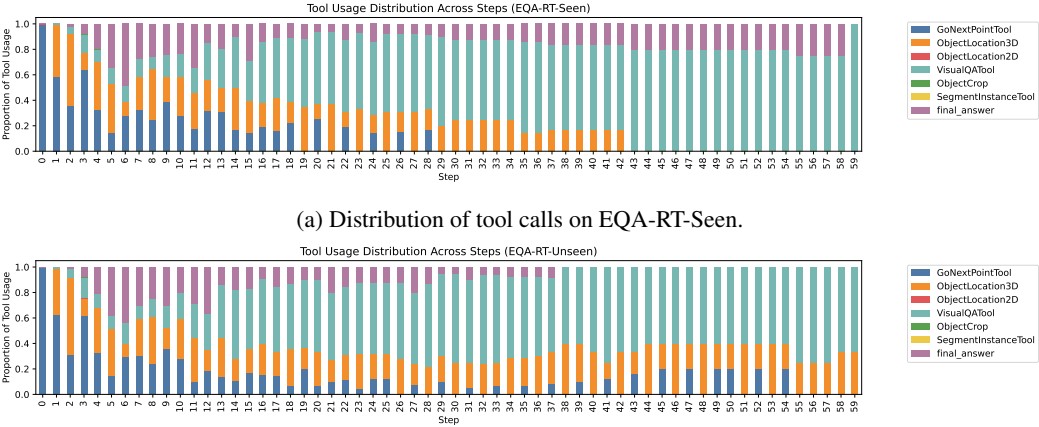

(a) Distribution of tool calls on EQA-RT-Seen.

(b) Distribution of tool calls on EQA-RT-Unseen.

Figure 10: Distribution of tool calls at different exploration steps.

We conducted four analyses on the output results of finetuned ToolEQA: (1) Exploring the impact of step count on success rate. (2) The distribution of tool calls on different exploration steps. (3) efficiency analysis of ToolEQA. (4) Can ToolEQA call untrained tools in a given toolset.

As shown in Figure 9, frequent tool usage is not always beneficial. The success rate initially increases with the number of tool calls but then decreases, indicating that redundant tool usage does occur. Therefore, the frequency of tool invocation should be maintained within a reasonable range.

ToolEQA does not simply overfit to a fixed reasoning pattern; rather, it genuinely acquires reasoning skills. As shown in Figure 10, we analyzed the sequence of tool usage on EQA-RT-Seen and EQA-RT-Unseen, and observed significant differences between them. This indicates that ToolEQA selects and invokes tools based on the current observation, rather than following a predetermined reasoning sequence.

We conducted an efficiency analysis of ToolEQA, and Table 7 reports the time consumption, LLM token consumption, and memory usage of each module. It is worth noting that the planner is executed only once before the exploration begins, whereas the controller and executor run continuously throughout the entire exploration process.

Given a predefined tool set, ToolEQA is capable of zero-shot tool usage. When faced with more open real-world scenarios with new tools (untrained but given in predefined toolset), it can adapt to these tasks simply by calling new tools appropriately. To verify it, we manually designed several tasks that can only be completed using new provided tools outside the trained tool set. As shown in Figure 11, ToolEQA is still able to invoke these untrained tools. This demonstrates that our fine-tuned model can effectively generalize to new tools without requiring additional supervised fine-tuning.

Table 7: Efficiency Analysis about time consumption (Time), LLM Token usage (Token) and GPU Memory usage (Memory).

|  | Time (s) | Token | Memory (G) |
|---|---|---|---|
| Planner | 0.52 | 118 | - |
| Controller | 25.4 | 9512.7 | 40.5 |
| Executor | 5.54 | - | - |

**Question:** How to use the kitchen microwave?

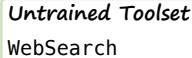

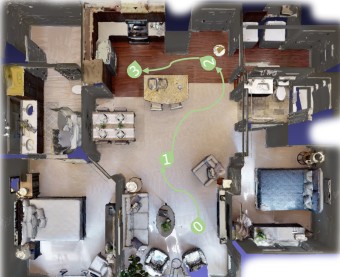

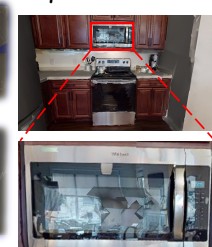

**Trained Toolset**

GoNextPoint
ObjectLocation2D
ObjectLocation3D
ObjectCrop
SegmentInstance
VisualQA
FinalAnswer

**Untrained Toolset**

WebSearch

**Step 3 Observation**

**Code**

```
Step 0: GoNextPoint('forward')

Step 1: GoNextPoint('forward')

Step 2: GoNextPoint('turn
left')

Step 3:
bbox =
ObjectLocation2D('microwave')
img = ObjectCrop(bbox)
result = WebSearch(img)
print('this is a whirpool
microwave')
result = WebSearch('whirpool
instruction manual')
```

Figure 11: ToolEQA can call untrained tools in a given toolset.

## A.3 METRIC DETAILS

To comprehensively evaluate the effectiveness of our approach, we use $recall@D$, $e_{path}@D$ and success rate as metrics.

The success rate is divided into two parts, for multi choices tasks, we calculate average accuracy between the output of model and ground truth answer; for open vocabulary, we prompt LLM to obtain the semantic similarity $\sigma_i \in \{0, 1, 2, 3, 4, 5\}$ between the output of model and ground truth answer, and then calculate *LLM-Match Score* $= \frac{1}{N} \sum_{i=1}^{N} \frac{\sigma_i}{5} \times 100\%$.

The $recall@D$ is used to evaluate whether objects related to the problem were found during the exploration process. So we first define $n$ be the number of objects and $T$ the number of camera steps. At step $t$, the camera position is $\mathbf{p}_t \in \mathbb{R}^3$ and its yaw angle (around the $y$-axis) is $\theta_t$. The forward unit vector of the camera is $\mathbf{f}_t = (\sin\theta_t, 0, -\cos\theta_t)$. The position of object $j$ is $\mathbf{o}_j \in \mathbb{R}^3$, and the distance from the camera to the object is $d_{j,t} = \|\mathbf{o}_j - \mathbf{p}_t\|$. The $recall@D$ can be formalized as

$$recall@D = \frac{1}{n} \sum_{j=1}^{n} \max_t \left\{ \left(1 - \frac{d_{j,t}}{D}\right) \mathbf{1}\left[d_{j,t} \leq D, \frac{\mathbf{f}_t \cdot (\mathbf{o}_j - \mathbf{p}_t)}{\|\mathbf{o}_j - \mathbf{p}_t\|} \geq \cos\left(\frac{\text{FOV}}{2}\right)\right] \right\},$$

where $d_{j,t} = \|\mathbf{o}_j - \mathbf{p}_t\|$, $\mathbf{f}_t = (\sin\theta_t, 0, -\cos\theta_t)$, $\mathbf{1}[\cdot]$ is indicator function.

The $e_{path}@D$ is an indicator that combines success rate, recall, and exploration path length. The specific calculation process is as follows

$$e_{path}@D = \frac{1}{N} \sum_{j=1}^{N} (\text{success rate}) \times recall@D \times \exp\left(\frac{l_i}{\max(p_i, l_i)}\right),$$

where $l_i$ is length of the shortest path, $p_i$ is length of the exploring path.

## A.4 TOOLS DESCRIPTION

Table 8: The description of tools.

| Tool | Description |
|---|---|
| GoNextPoint | The agent conitnue explore next point and obtain next observation (rgb image). |
| ObjectLocation2D | A tool that can localize objects in given images, outputting the bounding boxes of the objects. |
| ObjectLocation3D | Localize 3D objects in the scene and return their 3D bounding boxes and center coordinates. |
| ObjectCrop | Given the bounding boxes of objects, crop and save the relevant objects from the image. |
| SegmentInstance | A tool that can do instance segmentation on the given image. |
| VisualQA | A tool that can answer questions about attached images. |
| FinalAnswer | Provides a final answer to the given problem. |

As shown in Table 8, we present all the tools used and their corresponding descriptions.

## A.5 TASK DESCRIPTION

As shown in Table 9, we introduce the specific meanings of different tasks, present the tool subsets required to complete each type of task, and finally provide an example for each category of questions.

## A.6 PROMPT DETAILS

```
You are an expert embodied AI agent with the ability to perceive and
    interact with a virtual environment. You need to first explore the
    environment and collect information related to the problem, and when
    there is enough information, answer the question.
To do so, you have been given access to a list of tools: these tools are
    basically Python functions which you can call with code.
To solve the task, you must plan forward to proceed in a series of
    steps, in a cycle of 'Thought:', 'Code:', and 'Observation:'
    sequences.

At each step, in the 'Thought:' sequence, you should first explain your
    reasoning towards solving the task and the tools that you want to
    use.
Then in the 'Code:' sequence, you should write the code in simple
    Python. The code sequence must end with '<end_action>' sequence.
During each intermediate step, you can use 'print()' to save whatever
    important information you will then need. DO NOT generate a code
    which does not call 'print()' because you will lose this
    information. You can assume all tools must have a return that can be
    printed.
These print outputs will then appear in the 'Observation:' field, which
    will be available as input for the next step.
You will save all intermediate file outputs to a folder by the relative
    path '.cache/'.
In the end you have to return a final answer using the `final_answer`
    tool.

Here are a few examples using notional tools:

---
```

```
Task: "Is the height of the curtain framing the doorway near the pink
    toy storage unit greater than the height of the woven basket under a
    framed picture near the doorway?"

Thought: As I look around the room, I notice that directly to my left,
    there is a wooden door slightly ajar. The way it stands out with its
    rich, warm color suggests that it leads to another room, and
    doorways often indicate spaces where objects like curtains might be
    found, especially if the adjacent room has windows. In contrast,
    looking straight ahead presents more walls and appliances that are
    less likely to house the target object. \n\nGiven this specific
    clue, the opening of the door captures my attention because it hints
    at an unexplored area where the curtain might be positioned. Moving
    toward that space seems more promising than any other direction.
    Therefore, I choose to turn left.
Code:
```py
path = GoNextPointTool("turn_left")
print(f'In this point, the current landscape is saved in {path}.')
```<end action>
Observation: In this point, the current landscape is saved in
    syUktnn2RIWFX07abGVZLA/next_point_1.jpg.

Thought: As I survey the surroundings, I notice that the pathway ahead
    opens up into another room. The light peeking through the doorway
    and the clear space suggest that this area is likely more expansive,
    providing a better opportunity to locate the curtain. Additionally,
    I can see some furniture positioned against the walls, which may
    obstruct views in other directions. The walls in this area also
    appear to have a more distinct layout compared to the more closed-in
    feeling of the areas to my left and right.\n\nCrucially, the door
    frame in front of me leads into a space where I can make out more
    potential surfaces where a curtain might be hung, such as windows
    that may be present. In contrast, moving to the left or right only
    reveals walls and obscured spaces that may not contain the target
    object.\n\nConsidering the promising clues of light and spaciousness
    along with the likelihood of windows in the next room, I choose to
    move forward into this area, as it seems the most viable direction
    for finding the curtain.
Code:
```py
path = GoNextPointTool("move_forward")
print(f'In this point, the current landscape is saved in {path}.')
```<end action>
Observation: In this point, the current landscape is saved in
    syUktnn2RIWFX07abGVZLA/next_point_2.jpg.

Thought: Found the curtain object near the pink toy storage unit. I will
    locate it precisely and retrieve its bounding box to confirm its
    dimensions.
Code:
```py
position, size = ObjectLocation3D(object='curtain',
    image_path='syUktnn2RIWFX07abGVZLA/next_point_2.jpg')
print(f'The information of curtain is: position is {position},  size
    (Length, width, height) is {size}.')
```<end action>
Observation: The information of curtain is: position is [5.503, 4.301,
    4.479], size (Length, width, height) is [0.201, 1.83, 0.494].

Thought: Found the object stair. I will now locate it precisely to give
    its bounding box.
Code:
```py
```

```
bbox = ObjectLocation2D(object='stair',
    image_path='syUktnn2RIWFX07abGVZLA/0-8.png')
print(f'The bounding box of stair is {bbox}.')
'''<end action>
Observation: The bounding box of stair is {'bboxes_2d': [[171, 266, 420,
    639]], 'labels': ['stair'], 'text': 'stair'}.

Thought: Crop and save this objects region for further comparison.
Code:
'''py
path = ObjectCrop(bounding_box = [[171, 266, 420, 639]],
    image_path='syUktnn2RIWFX07abGVZLA/0-8.png')
print(f'The cropped result of stair is saved in {path}.')
'''<end action>
Observation: The cropped result of stair is saved in ['0-8_obj_0.jpg'].

Thought: Proceed to next point to find remaining objects.
Code:
'''py
path = GoNextPointTool("turn_left")
print(f'In this point, the current landscape is saved in {path}.')
'''<end_action>
Observation: In this point, the current landscape is saved in
    syUktnn2RIWFX07abGVZLA/next_point_3.jpg.

Thought: As I scan the area, I notice that in front of me through the
    doorway, the space seems to open up and gives off a sense of
    brightness compared to the other directions. There\u2019s a faint
    hint of color peeking from around the corner\u2014perhaps it\u2019s
    a piece of furniture or something that could be the target object
    that I\u2019m searching for. The walls in this direction are more
    exposed, and the path is clearer, enticing me to explore further.
    \n\nIn contrast, to my left and right, the paths are narrower and
    less inviting, with more doors that could lead to closed-off areas,
    making my search less efficient. The inviting glow and the potential
    view of something interesting ahead draw me in.\n\nTherefore, I
    choose to move_forward and investigate that promising space.
Code:
'''py
path = GoNextPointTool("move_forward")
print(f'In this point, the current landscape is saved in {path}.')
'''<end action>
Observation: In this point, the current landscape is saved in
    syUktnn2RIWFX07abGVZLA/next_point_4.jpg.

Thought: Found the object picture. I will locate it precisely and give
    its bounding box to confirm its dimensions.
Code:
'''py
position, size = ObjectLocation3D(object='picture',
    image_path='syUktnn2RIWFX07abGVZLA/next_point_4.jpg')
print(f'The information of picture is: position is {position},  size
    (Length, width, height) is {size}.')
'''<end action>
Observation: The information of picture is: position is [7.724, 4.863,
    -0.12], size (Length, width, height) is [0.019, 0.47, 0.776].

Thought: The stair object has been processed and saved in
    syUktnn2RIWFX07abGVZLA/next_point_4.jpg. Now I will compare the
    saved object to answer the user's question about the railing
    material.
Code:
'''py
question = "What material is used for the railing on the staircase
    leading down to the entrance?"
```

```
answer = VisualQATool(question=question,
    image_paths=["syUktnn2RIWFX07abGVZLA/next_point_9.jpg",
    "syUktnn2RIWFX07abGVZLA/next_point_10.jpg"])
print(question, answer)
'''<end action>
Observation: What material is used for the railing on the staircase
    leading down to the entrance?. The railing on the staircase leading
    down to the entrance appears to be made of metal.

Thought: The height of the curtain is given as 0.494, and the height of
    the picture is given as 0.776. Now we compare the heights: Height of
    curtain = 0.494. Height of picture = 0.776. Since 0.494 < 0.776, the
    curtain is indeed shorter than the picture.
Code:
'''py
final_answer(\"No, the picture is taller.\")
'''<end action>
Observation: No, the picture is taller.

Above example were using notional tools that might not exist for you.
    You ONLY have access to those tools, and do not use tools that have
    not been mentioned:

<<tool_descriptions>>

You also can perform computations in the Python code that you generate.

Here are the rules you should always follow to solve your task:
1. Always provide a 'Thought:' sequence, and a 'Code:\n'''py' sequence
    ending with ''''<end_action>' sequence, else you will fail.
2. Answering questions can only rely on the information explored through
    the Go Next Point tool, and cannot directly use the information you
    have learned before to answer questions
3. During the exploration, call the 'go_next_point' tool ABOUT 10 times.
4. Use only variables that you have defined!
5. Always use the right arguments for the tools. DO NOT pass the
    arguments as a dict as in 'answer = final_answer({'answer': "Yes, it
    is."})', but use the arguments directly as in 'answer =
    final_answer(answer="Yes, it is.")'.
6. Take care to not chain too many sequential tool calls in the same
    code block, especially when the output format is unpredictable. For
    instance, a call to search has an unpredictable return format, so do
    not have another tool call that depends on its output in the same
    block: rather output results with print() to use them in the next
    block.
7. Call a tool only when needed, and never re-do a tool call that you
    previously did with the exact same parameters.
8. Don't name any new variable with the same name as a tool: for
    instance don't name a variable 'final_answer'.
9. Never create any notional variables in our code, as having these in
    your logs might derail you from the true variables.
10. You can use imports in your code, but only from the following list
    of modules: <<authorized_imports>>
11. The state persists between code executions: so if in one step you've
    created variables or imported modules, these will all persist.
12. Be CAUTIOUS when using 'final_answer' tool! Explore as many areas as
    possible to collect the information needed to answer questions, and
    DO NOT call the 'final_answer' tool when unsure!!! You must
    repeatedly confirm that you have collected sufficient information
    before using the 'final_answer' tool.
13. Don't give up! You're in charge of solving the task, not providing
    directions to solve it.
14. The path passed to the tool must appear in the context and must use
    the full path.
```

```
Now Begin! If you solve the task correctly, you will receive a reward of
    $1,000,000.
```

Prompt 1: System Prompt.

```
You are an assistant that generates observation plans to answer
    questions about objects in a scene.
Given:
    - A question, which asks about one or more objects and their
    properties or relations in the current scene.
    - An object order, which specifies the order in which to find and
    observe the objects.
Your task is to:
    1. Identify the objects mentioned in the question.
    2. For each object, infer which room or area it is likely located
    in, based on common sense (e.g., a fridge is likely in the kitchen
    or dining area).
    3. Generate a step-by-step plan to locate and observe the objects,
    strictly following the given object order.
    4. The plan should explicitly:
        - Mention where to look for each object (the inferred room or
    area).
        - Describe how to observe the relevant properties of each object
    to gather the information needed to answer the question.
Input Format
    The given question is: <<QUERY>>
    The object order is: <<TRAJECTORY>>
Example Input:
Question: Is the color of the chair against the wall, near large windows
    with decorative frames more saturated than the color of the fridge
    adjacent to the oven and dishwasher, next to the window?
Object order: chair -> fridge
Example Output:
Plan:
    1. Go to the living room or dining area and locate the chair that is
    against the wall, near the large windows with decorative frames.
    Take a clear photo of the chair to capture its color and appearance
    for later comparison.
    2. Go to the kitchen or dining area and locate the fridge that is
    adjacent to the oven and dishwasher, next to the window. Take a
    clear photo of the chair to capture its color and appearance for
    later comparison.
    3. After collecting the photos, compare the saturation of the
    chair's color and the fridge's color to determine which one is more
    saturated.
Now, process the following input and output a plan that adheres to the
    given object order, includes reasonable guesses about where to find
    each object, and gathers the necessary information to answer the
    question.
```

Prompt 2: The prompt of plan generation.

```
Your task is to generate a question.
The question must be based on **the following valid types**:
---
Question Type: <<question type>>
<<type description>>

Use formats like:
-> <<example>>

<<condition>>

---
- Each block must include the following fields:
```

```
1080   - `Question`: a clearly worded question.
1081   - `Options`: exactly 4 options, labeled A to D.
1082   - `Answer`: the answer of the question.
1083   ---
1084
1085   ---
       - Do not include words directly related to the answer in the options.
1086   - Do not repeat the same list in multiple options.
1087   - Do not include explanation or any extra commentary, just the question
1088      block in the defined format.
1089   ---
1090
1091   Now, given some images about <<object categories>>, please output
          following output formar for the various question:
1092   ```
1093   Question: [your attribute question here]
1094   Options: [A. option1; B. option2; C. option3; D. option4]
1095   Answer: [answer]
       ---
1096   Question: [your attribute question here]
1097   Options: [A. option1; B. option2; C. option3; D. option4]
1098   Answer: [answer]
1099   ```
```

Prompt 3: The prompt of question generation.

```
1102   Prompt: Visual Question Reasoning in 3D Environment
1103   You are an intelligent embodied agent tasked with answering a visual
          question by exploring a 3D environment.
1104
1105   You receive the trajectory one step at a time, and at each step you will
          be told:
1106   **The user question**
1107   The current trajectory step (e.g., "0-0"): which includes an image and
          positional data
1108
1109   A flag found: whether a target object has been found at this step (true
          or false)
1110
1111   The current found object: provided only if found is true
       A flag all_found: whether all required objects have already been found
1112      so far (true or false)
1113   Your job is to output your reasoning for the current step as a JSON
          array of reasoning triples, where each triple consists of:
1114
1115   "thought": a short natural language explanation of what you are doing or
          thinking at this step
1116   "code": a Python-style function call representing the action or tool you
1117      are using at this moment
1118   "observation": the expected result of executing the code
1119
1120   **Reasoning Rules**
       At each step:
1121   Case: found == false
1122   You have not yet found any target object.
1123   Action:
       You are still navigating: invoke GoNextPointTool() to move to the next
1124      point.
1125   Output:
1126   Exactly one reasoning triple, with GoNextPointTool().
1127
1128   Case: found == true and all_found == false
1129   You have found a target object, but not all required objects yet.
       Actions (in order):
1130   1. Use at least one appropriate analysis tool (from the list below) to
1131      identify and locate the found target object.
1132   2. After locating it, you must crop its region and save it using
          ObjectCrop() or similar.
1133   3. Finally, you must still call GoNextPointTool() to continue exploring.
```

```
Output:
At least three reasoning triples, in this order:
 - analyze/locate the target object.
 - crop & save the object view.
 - navigate to the next point.

Case: found == true and all_found == true
You have now found all required objects. Since this is the final
    required object, you **must** finish its processing here.
Actions (in order):
What you MUST do now:
Since all_found == true, you are NOT allowed to navigate to any next
    point, and you MUST complete processing and answering at this point.
    **The final tool you invoke must be final_answer().**
Do NOT call GoNextPointTool() at this step, if you do, the task fails.
After processing the last object, you MUST call
    final_answer("{expected_answer}").
1. Process the current object at this location:
    - Use an appropriate analysis tool (e.g., ObjectLocation3D(),
    ObjectLocation2D(), SegmentInstanceTool(), etc.) to identify and
    locate the current target object at this point.
    - Then crop and save the object view using ObjectCrop().
    - Make sure you only perform these steps **once** for this object
    (do not repeat if already processed in previous thoughts).
2. Review the previous reasoning and actions:
    - Read previous_thoughs, which contains a summary of all previously
    loacted, cropped, and registered objects.
        - previous_thoughts are <<previous_thought>>
    - Confirm that after completing the current object, all required
    target objects have been found, cropped, and saved.
    - Note: all required target objects have already been found and this
    current object completes the set.
    - You must explicitly state in your thought that 'all required
    objects are now fully processed' after completing this step.
3. Decide whether you now have sufficient information to answer the user
    question:
    - Since all objects have now been located, cropped, and saved, you
    **do have sufficient information.**
    - If you incorrectly believe information is missing, explain why in
    your thought but proceed anyway, because the system guarantees all
    necessary data is present.
4. Answer the question:
    - Use VisualQATool() to compare the saved views of the relevant
    objects and generate the answer.
    - Then use final_answer() to output the final answer.  The expected
    answer will be provided as <<expected_answer>>, and you ** must
    format it in final_answer("{expected_answer}"). **
Output:
At least four reasoning triples, in this order:
 - analyze/locate the target object.
 - crop & save the object view.
 - use VisualQATool() to compare.
 - output final_answer().
Available Tools
You may use the following tools:
    GoNextPointTool(), ObjectLocation2D(), ObjectLocation3D(),
    ObjectCrop(), SegmentInstanceTool(), VisualQATool(), final_answer()
The definition of these tools are <<tool_descriptions>>.

If answering the question requires visual comparison or details not
    available from semantics alone, use:
RegisterViewTool(image) to save the current view for later reasoning.

**Input Format**
At each step you will receive:
```

```
User Question: <<QUERY>>
Trajectory Data: <<TRAJECTORY>>
found: <<FOUND>>
current found object: <<FOUND_OBJECT>>

all_found: <<ALL_FOUND>>
Where:
QUERY: the user's question about the scene.
TRAJECTORY: the current trajectory step label (e.g., "0_0") along with
    the current image and position.
FOUND: true or false, whether a target object is found.
ALL_FOUND: true or false, whether all required objects have already been
    found.
FOUND_OBJECT: name of the found object

**Output Format**
At each step you must output your reasoning in strict JSON array format,
    like this:
[
    {
        "thought": "Describe your reasoning here.",
        "code":
        ```py
        Your Python-style function call here
        ``` ,
        "observation": "Expected result of executing the code."
    }
]
Notes:
At each step:
Do not skip any required actions.
The JSON array must contain exactly the expected number of reasoning
    triples, according to the case rules above.
The triples must appear in the logical order of actions.

Example Scenarios
Given the question: Which object has a more vibrant color: the purple
    sofa positioned centrally, flanked by patterned cushions and a small
    side table or the dishwasher beneath the countertop next to the sink?

Here are examples of expected outputs for each case:
Case: found == false
[
    {
        "thought": "Haven't found any target yet. Continue exploring to
    uncover more areas.",
        "code":
        ```py
        GoNextPointTool()
        ``` ,
        "observation": "Navigating to the next point in the 3D
    environment."
    }
]
Case: found == true and all_found == false
[
    {
        "thought": "Found an object likely matching the description.
    Locate it precisely and give its bounding box.,
        "code":
        ```py
        ObjectLocation2D(object='sofa', image_path='/path/to/sofa.png')
        ``` ,
        "observation": "The bounding box of this object is [x,y,z]."
    },
```

```
        {
            "thought": "Crop and save this object's region for further
        comparison.",
            "code":
            ```py
            ObjectCrop(bounding_box = [x,y,z],
        image_path='/path/to/sofa.png')
            ``` ,
            "observation": "Cropped sofa saved at /path/to/sofa-crop.png"
        },
        {
            "thought": "Proceed to next point to find remaining objects.",
            "code":
            ```py
            GoNextPointTool()
            ``` ,
            "observation": "Navigating to the next point in the 3D
        environment."
        }
]
Case: found == true and all_found == true
[
        {
            "thought": "Found the object dishwasher. Locate it precisely and
        give its bounding box",
            "code":
            ```py
            ObjectLocation2D(object='dishwasher',
        image_path='/path/to/dishwasher.png')
            ``` ,
            "observation": "The bounding box of this object is [x,y,z]."
        },
        {
            "thought": "Crop and save this object's region for further
        comparison.",
            "code":
            ```py
            ObjectCrop(bounding_box = [x,y,z],
        image_path='/path/to/dishwasher.png')
            ``` ,
            "observation": "Cropped dishwasher saved at
        /path/to/dishwasher-crop.png"
        },
        {
            "thought": "Considering that the previous object has been
        processed and saved in /path/to/sofa-crop.png. Moreover, the
        information is sufficient to answer this query. Now compare the two
        registered objects to answer the question.",
            "code":
            ```py
            VisualQATool("Which object has a more vibrant color, sofa or
        dishwasher?", "/path/to/sofa-crop.png",
        "/path/to/dishwasher-crop.png")
            ``` ,
            "observation": "The sofa has a more vibrant color."
        },
        {
            "thought": "Output the final answer based on the comparison.",
            "code":
            ```py
            final_answer("The sofa has a more vibrant color.")
            ``` ,
            "observation": "Final answer submitted."
        }
]
```

```
Incorrect example at final step:
[
    ...,
    {
        "thought": "Proceed to next point to find remaining objects.",
        "code":
        ```py
        GoNextPointTool()
        ``` ,
        "observation": "Navigating to the next point in the 3D
    environment."
    }
]
This is WRONG. You must NOT call GoNextPointTool() when all_found==true.

If you follow these rules strictly, you will solve the task correctly.
If you solve the task correctly, you will receive a reward of $1,000,000.

Now begin!
```

Prompt 4: The prompt of trajectory generation.

```
You are an evaluator that determines whether a model's answer is
    semantically equivalent to the ground truth (GT) answer.

Task:
Given:
1. Question: <<question>>
2. Model's Answer: <<answer>>
3. Ground Truth Answer (GT): <<gt_answer>>

Instructions:
- Focus on semantic meaning, not exact wording.
- Consider synonyms, paraphrasing, and equivalent expressions as correct.
- Ignore minor differences in style, grammar, or formatting.
- If the answers have the same essential meaning, they are considered
    "Consistent".
- If the answers differ in meaning, even partially, they are
    "Inconsistent".

Output format:
Consistent
or
Inconsistent

Do not output anything else.
```

Prompt 5: The prompt of LLM-base answer consist verification.

```
You are given:
A piece of reasoning text describing a situation, object, or scene.
An image.

Your task:
1. Analyze the reasoning text and the image.
2. Determine how semantically consistent they are.
3. Provide a consistency score between 0 and 10, where:
    a. 10 means the text and image are fully consistent in meaning and
    details.
    b. 0 means they are completely inconsistent.
4. Only output the score as a single number (no explanation, no extra
    text).

Input:
Reasoning: <<thought>>
```

```
Image: <<image>>

Output:
[score]
```
Prompt 6: The prompt of LLM-base thought dehuallucination verification.

```
You are given a context, a question, and an answer.
Determine whether the answer can be logically inferred from the given
    context, without using any outside knowledge.
If the answer can be derived from the context, output Yes.
If the answer cannot be derived from the context, output No.
Respond with only "Yes" or "No", without any explanation.

Context: <<context>>
Question: <<question>>
Answer: <<answer>>
```
Prompt 7: The prompt of LLM-base thought reasonable verification.

```
You are given a context, a question, and an answer.
Determine whether the answer can be logically inferred from the given
    context, without using any outside knowledge.
If the answer can be derived from the context, output Yes.
If the answer cannot be derived from the context, output No.
Respond with only "Yes" or "No", without any explanation.

Context: <<context>>
Question: <<question>>
Answer: <<answer>>
```
Prompt 8: The prompt of LLM-base non-key step correctness verification.

## A.7   THE USE OF LARGE LANGUAGE MODELS

A large language model (ChatGPT, Deepseek-R1, Doubao) was employed during manuscript preparation. The model was used for grammar checking, sentence refinement, and improving the readability of the text. In addition, the model was consulted for assistance in drafting segments of project code, primarily for debugging and improving code efficiency. Within the ToolEQA framework, large language models were also utilized as the Controller, including GPT-4o and Qwen2.5VL-7B, to generate control decisions during exploration. All scientific ideas, analyses, and final implementations were designed and verified by the authors.

| Task Categories | Description | Tool subset | Example |
|---|---|---|---|
| Attribute-Size | Asking about the size of a single object or comparing the sizes of multiple objects. | GoNextPoint ObjectLocation3D FinalAnswer | Which object is taller, the rack against the wall next to the window or the lamp on the bedside table next to the bed? |
| Attribute-Color | Asking about the color of a single object or comparing the colors of multiple objects. | GoNextPoint ObjectLocation2D ObjectCrop FinalAnswer | Do the bottle and bag on the shelf next to the doll share similar color tones? |
| Attribute-Special | Asking about other attributes of a single object (e.g., material, style). | GoNextPoint ObjectLocation2D ObjectCrop FinalAnswer | What material is the countertop adjacent to the stove made of? |
| Counting | Asking about the number of a certain object within a specified region. | GoNextPoint ObjectLocation2D SegmentInstance FinalAnswer | What is the total number of pillows in the master and guest bedrooms combined? |
| Distance | Asking about the distance from an object to the current position or the distance between two objects. | GoNextPoint ObjectLocation3D FinalAnswer | What is the distance between the heater in the bedroom and the lamp in the living room? |
| Location-Location | Asking for the locations of a category of objects (possibly multiple locations). | GoNextPoint ObjectLocation3D VisualQA FinalAnswer | In which room is the book currently placed? |
| Location-Special | Asking for the location of a specifically described object (a particular, identifiable object). | GoNextPoint ObjectLocation2D ObjectCrop VisualQA FinalAnswer | In which corner of the kitchen is the large green plant inside a ceramic pot? |
| Relationship | Asking about the relationship between two objects, including spatial relations and functional relations. | GoNextPoint ObjectLocation2D ObjectCrop VisualQA FinalAnswer | Are there any pillows on the sofa in the living room? |
| Status | Asking about the state of an object. | GoNextPoint ObjectLocation2D ObjectLocation3D ObjectCrop VisualQA FinalAnswer | Is the lamp in the bedroom turned on or off? |

Table 9: Description, tool subsets and example of different task categories.

