# OpenReview forum: "Multi-Step Reasoning for Embodied Question Answering via Tool Augmentation"
_ICLR.cc/2026/Conference — Submitted to ICLR 2026_

### Official Review · Reviewer_mden · 2025-11-01

**Soundness:** 3
**Presentation:** 3
**Contribution:** 3
**Rating:** 8
**Confidence:** 3

**Summary:**

The paper proposes ToolEQA, a novel method for embodied question answer (EQA). The agent itself utilizes multistep reasoning and uses tools and code execution as the means to interaction with the virtual environment. In addition to that, the paper synthetically generated a EQA dataset (EQA-RT), by 1) asking GPT-4o to generate question-answer pairs, 2) generating optimal trajectory with A* and enriching each step with reasoning and tools, and 3) verifying each task. The ToolEQA demonstrates improved performance on EQA benchmarks and in particular finetuning clearly boosts performance.

**Strengths:**

- The paper proposes a simple yet effective method for EQA.
- The data generation pipeline and the dataset is very useful. There is clear performance gain from finetuning.
- The paper conducted comprehensive experiments.

**Weaknesses:**

- There could be more ablation of the method to better understands the contribution of each component.

**Questions:**

- How are the set of tools defined? Among the code blocks, are there any interesting action other than `print`?

---

> ### Author Response · Authors · 2025-11-24
>
> >**Q1:** There could be more ablation of the method to better understands the contribution of each component.
>
> **A:** Thanks for your suggestions. We conduct an ablation study on EQA-RT-Unseen to examine the roles of thought, the planner, and tool calls. Results are shown in Table 1, demonstrating that thought and tool are important for performance, and combine thought with tool calls yields the optimal performance. Without tools, the VLM directly outputs exploration directions, preventing the agent from acquiring critical information for certain question types (e.g., size comparisons) and reducing success rates. Without thought, the model outputs tool calls without reasoning, which provides necessary information but leaves exploration unguided and inefficient. Omitting the planner removes global guidance at the start of exploration, resulting in longer paths. Together, thought, tool calls, and the planner achieve optimal performance, guiding exploration while ensuring the agent obtains essential information for answering questions. The analysis has been added to the revised manuscript.
>
> **Table 1: Abaltion on thought, tool and planner.**
> | Model | Succ. $\uparrow$ | recall@10 $\uparrow$ | e_path@10 $\uparrow$ | length $\downarrow$ |
> | - | - | - | - | - |
> | ExploreEQA              | 31.4 | 0.09 | 0.10 | 65.15 |
> | ToolEQA (w/o thought)   | 48.6 | 0.11 | 0.19 | 30.52 |
> | ToolEQA (w/o tool)      | 41.9 | 0.15 | 0.17 | 23.61 |
> | ToolEQA (w/o planner)   | 55.2 | 0.16 | 0.23 | 25.97 |
> | ToolEQA                 | 56.1 | 0.17 | 0.24 | 16.63 |
>
> >**Q2:** How are the set of tools defined? Among the code blocks, are there any interesting action other than print?
>
> **A:** Table 8 in the appendix describes our tool definitions, and the implementation details of the tool is shown in the Table 2.
>
> **Table 2: Details of tool implementation.**
> | Tool | Implement |
> | - | - |
> | GoNextPoint| We reconstruct the 3D scene into a voxel representation and project the voxels onto a 2D map. Based on this map, we employ a frontier-based exploration strategy to obtain multiple candidate exploration directions, and the final direction is selected according to the input parameters. |
> |ObjectLocation2D|[GroundingDino](https://github.com/IDEA-Research/GroundingDINO)|
> |ObjectLocation3D|[DetAny3D](https://github.com/OpenDriveLab/DetAny3D)|
> |ObjectCrop|Python|
> |SegmentInstance|[SAM2](https://github.com/facebookresearch/sam2)|
> |VisualQA|[Qwen2.5VL](https://github.com/QwenLM/Qwen3-VL)|
> |FinalAnswer|[Qwen2.5VL](https://github.com/QwenLM/Qwen3-VL)|
>
> Moreover, tool usage is not limited to print operations but also involves logical reasoning. For example, given the question: ‘Which object is taller, the chair at the corner of the dining area next to the marble table or the kitchen cabinet adjacent to the stainless steel refrigerator and facing the dining table?’, some of the tool-invocation steps are as follows:
> ```python
> # Step 2:
> position, size = ObjectLocation3D(object='kitchen cabinet', image_path='next_point_2.jpg')
> print(f'The information of kitchen cabinet is: position is {position},  size (Length, width, height) is {size}.')
>
> # Step 4:
> position, size = ObjectLocation3D(object='chair', image_path='next_point_4.jpg')
> print(f'The information of chair is: position is {position},  size (Length, width, height) is {size}.')
>
> # Step 5:
> chair_height = 0.93
> cabinet_height = 0.44
> if chair_height > cabinet_height:
>     answer = "A. The chair is taller"
> elif cabinet_height > chair_height:
>     answer = "B. The kitchen cabinet is taller"
> else:
>     answer = "C. They are the same height"
> print(answer)
> ```
> This indicates that our code blocks not only call tools, but also have the ability to autonomously process information obtained through tools (such as size comparisons), demonstrating the flexibility of our method.
> In future work, we may incorporate tools related to environmental interaction, such as robotic arm manipulation and grasping.

---

> ### Comment · Reviewer_mden · 2025-11-25
> **Post-Rebuttal Summary**
>
> The authors addressed the issue I raised on the lack of ablation. I do not have outstanding concerns.
>
> I do not dispute the fact that augmenting external tools for EQA is a marginal contribution. I think the main contribution lies in EQA-RT, the automatic data generation pipeline and the dataset.

---

### Official Review · Reviewer_zT4c · 2025-11-01

**Soundness:** 2
**Presentation:** 3
**Contribution:** 2
**Rating:** 2
**Confidence:** 4

**Summary:**

The paper contributes a framework, ToolEQA that augments VLM-based EQA agents with tools (eg. bounding box detection, navigation primitives) for eliciting multi-step reasoning. The work further develops a data generation pipeline involving task creation, trajectory generation, reasoning trace generation and data verification, collecting a EQA-RT dataset. The results demonstrate that ToolEQA agents outperform prior works on the proposed EQA-RT benchmark and EXPRESS-Bench framework over most prior works, and that finetuning base VLM models on the collected dataset further improves performances.

**Strengths:**

1. Authors explore the relatively understudied extension of tool usage to the application of embodied question answering.
2. Usefulness of generated data: finetuning on the collected dataset improves tool usage and EQA success rates.
3. The paper's ToolEQA agent outperforms prior method (Fine-EQA) on Fine-EQA's benchmark (EXPRESSBench).

**Weaknesses:**

1. The paper has critical gaps in evaluation and analysis that do not fully establish the benefits of tool usage as well as improvements over prior works:

    a. The authors do not compare to a comparable baseline that follows the same pipeline but replaces the tools with VLM (eg. use the same VLM to compare sizes of objects in two frames). It is not clear if the gains are coming from breaking down reasoning into tools (multi-step reasoning) or the use of external tools.

    b. The evaluation metric ($e_{\text{path}}$) combines recall, and hence it is not clear if ToolEQA, and its finetuning, result in efficiency improvements. Can the authors report the efficiency metric from OpenEQA?

    c. In tables 4 and 5, it is not clear if the ToolEQA used in evaluations is the finetuned version, leaving open the following questions:  Does finetuning on EQA-RT improve performance on other benchmarks? Are the performance improvements over prior works (eg.  Fine-EQA) coming from additional finetuning?

    d. GraphEQA, which also incorporates multi-step reasoning, appears to be better than HM-EQA on the only evaluation in Table 5. In the absence of additional comparisons and ablations, it is not clear if the components introduced in this work are essential.

    e. The method does not maintain and have access to a global scene representation (eg. scene graph, semantic map), and thus makes locally optimal decisions (eg. the doorway that looks promising without consideration of the expected scene layout). It is not clear how this choice affects the results and the comparisons to prior works (eg. Fine-EQA, GraphEQA) that maintain such representations.

    f. The work invokes a VLM in every step for low-level navigation commands ("turn left"), compared to prior works (eg. GraphEQA) that predict navigation targets offloading navigation to low-level planner. The effect of this choice on results is also unclear.

    g. While the work includes and additional test set with unseen scenes, it is not clear if the reported generalization (similar performance on seen and unseen scenes) is due to the unseen scenes having fewer target objects (Fig. 7c). It might be useful to break down the performance on the number of target objects to see if it has an effect on the performance of different splits (as hypothesized in the paper in L310).

2. Multi-step reasoning with tool usage has been explored to great extent in non-embodied settings (ViperGPT, VisProg, T3-Agent) and multi-step reasoning has been explored for embodied question answering (Fine-EQA). As a result the technical contributions of the work are limited, i.e., applying T3-Agent to embodied QA.
3. Details of critical components of the method (eg. planner, tool implementations, prompts used) are missing in the main paper and the supplementary. These are described under "Questions".

**Questions:**

1. The descriptions of tools are missing. For example, it is not clear how VisualQA or SegmentInstance are implemented and used.
2. It would be useful to include the prompts used for data generation and the VLM-based controller.
3. Have the authors considered using the Exploration-Answer Consistency (EAC) metric from FineEQA? Why do authors not use a consistent set of metrics cross all benchmarks?
4. L175: Details of the planner are missing. What structure do the generated plans follow ($p$ in L203)? The planner also seems to be absent in the presented example.
5. Can authors include more details and examples for the different categories of tasks (location-location, location-special)?
6. It would be useful to include prompts for the controller and the data generation pipeline.
7. For EQA task verification, why is it important to use both object detector and LLM? How do these compare?
8. How efficient is the data generation procedure? What fraction of trajectories get filtered out for each of the different reasons (L283-L286)?
9. L320: It is not clear how finetuning to predict the final answer ($c_t = \texttt{FinalAnswer}$) would hurt the model? Can the authors elaborate more on how this encourages the model to use its biases by ignoring the context?
10. The length of trajectories is lower than the trajectories -- an average of 12.69 (L321). Based on the distances in Table 6, does this mean the agent moves by more than 1m in each step? How does this not prevent from taking finer steps for moving closer to objects or navigating cluttered scenes?
11. Some typos and questions in writing:
- L119: ~REALTED~ RELATED
- L131: ~the lack of~ they lack
- L156: ~finetunes~ finetuned
- L183: ~executing~ executes
- L184: By invoking the executor to: Typo or incomplete
- L171: The agent should not have access to an additional scene representation $S$ beyond the observations $o_i$?
- L361: ~we~ We

---

> ### Author Response · Authors · 2025-11-24
>
> >**W1a:** The authors do not compare to a comparable baseline that follows the same pipeline but replaces the tools with VLM (eg. use the same VLM to compare sizes of objects in two frames). It is not clear if the gains are coming from breaking down reasoning into tools (multi-step reasoning) or the use of external tools.
>
> **R:** We conduct an ablation study to evaluate the roles of thought and tool calls. In the w/o tool setting, the VLM directly outputs the exploration direction in place of tools, whereas in the w/o thought setting, the model directly outputs the tool calls without reasoning. The results in Table.1 show that thought and tool are important for performance, and combine thought with tool calls yields the optimal performance. Thought provides guidance for determining the exploration direction, however, the lack of tools prevents the agent from obtaining key information for certain categories of questions (e.g., size comparison), leading to reduced success rates. Conversely, incorporating tools provides essential information for answering questions, but without thought to interpret the scene and historical context, the exploration direction becomes unguided, ultimately reducing exploration efficiency.
>
> **Table 1: Abaltion on thought and tool.**
> | Model | Succ. $\uparrow$ | recall@10 $\uparrow$ | e_path@10 $\uparrow$ | length $\downarrow$ |
> | - | - | - | - | - |
> | ExploreEQA            | 31.4 | 0.09 | 0.10 | 65.15 |
> | ToolEQA (w/o thought) | 48.6 | 0.11 | 0.19 | 30.52 |
> | ToolEQA (w/o tool)    | 41.9 | 0.15 | 0.17 | 23.61 |
> | ToolEQA (w/o planner) | 55.2 | 0.16 | 0.23 | 25.97 |
> | ToolEQA               | **56.1** | **0.17** | **0.24** | **16.63** |
>
> >**W1b:** The evaluation metric combines recall, and hence it is not clear if ToolEQA, and its finetuning, result in efficiency improvements. Can the authors report the efficiency metric from OpenEQA?
>
> **R:** We have updated the efficiency metrics exploration distance on the OpenEQA dataset in the Table 5 of manuscript. As shown in Table.2, combined with the comparisons against baselines and the experiments on the HM-EQA dataset and ExpressBench, the results show that ToolEQA substantially shortened the exploration distance, achieving dramatically shorter exploration trajectories while maintaining comparable success rates.
>
> **Table 2: QA-Agent performance on existed benchmarks.**
> | Model | HM-EQA (Succ.) | HM-EQA (L) | OpenEQA (Succ.)| OpenEQA (L) |
> | - | - | - | - | - |
> | Explore-EQA | 51.5 | 38.87 | 28.3 | 25.45 |
> | Efficient-EQA | 54.3 | 30.16 | - | - |
> | Memory-EQA | 63.4 | 33.54 | 34.6 | 11.32 |
> | Fine-EQA | 53.3 | 34.75 | 29.4 | 13.77 |
> | Graph-EQA | 63.5 | - | 30.1 | 11.96 |
> | ToolEQA (0-shot) | 61.0 | 18.98 | 33.1 | 8.38 |
> | ToolEQA (ft) | **62.3** | **18.26** | **35.5** | **6.96** |
>
> >**W1c:** In tables 4 and 5, it is not clear if the ToolEQA used in evaluations is the finetuned version, leaving open the following questions: Does finetuning on EQA-RT improve performance on other benchmarks? Are the performance improvements over prior works (eg. Fine-EQA) coming from additional finetuning?
>
> **R:** In both Table 4 and Table 5 of the manuscript, we use the fine-tuned version of ToolEQA, demonstrating that fine-tuning indeed improves performance on other benchmark OpenEQA, HM-EQA and ExpressBench. As shown in Table 3, we additionally include the zero-shot results in these datasets. The comparison shows that ToolEQA already yields noticeable gains across benchmarks in the zero-shot setting, and achieves further improvements after supervised fine-tuning.
>
> **Table 3: Comparison between the zero-shot model and the finetuned model on different benchmark.**
> | Model | OpenEQA (succ.) | HM-EQA (succ.) | ExpressBench (succ.) |
> | - | - | - | - |
> | ToolEQA (zero-shot)   | 33.1 | 61.0 | 40.65 |
> | ToolEQA (finetune) | 35.5 | 62.3 | 42.21 |
>
> We have updated the more detailed metrics in tables 3, 4, and 5 of the manuscript.
>
> >**W1d:** GraphEQA, which also incorporates multi-step reasoning, appears to be better than HM-EQA on the only evaluation in Table 5. In the absence of additional comparisons and ablations, it is not clear if the components introduced in this work are essential.
>
> **R:** HM-EQA cannot comprehensively evaluate an agent’s capabilities, so the results on this dataset cannot be used as direct evidence for whether our thought, code, and planner frameworks are necessary. Specifically, the questions in the HM-EQA dataset require only a single observation to obtain all the information needed to answer them, and the multiple-choice options provide additional priors. Under such conditions, ToolEQA’s long-range reasoning abilities cannot be fully utilized. Moreover, as shown in Table 1 in the response of W1a, we have demonstrated the importance of thought, code, and planner.

---

> ### Author Response · Authors · 2025-11-24
>
> >**W1e:** The method does not maintain and have access to a global scene representation (eg. scene graph, semantic map), and thus makes locally optimal decisions (eg. the doorway that looks promising without consideration of the expected scene layout). It is not clear how this choice affects the results and the comparisons to prior works (eg. Fine-EQA, GraphEQA) that maintain such representations.
>
> **R:** We also maintain a global scene representation. Within the GoNextPoint tool, we construct a voxel-based global map, on which candidate frontier points are generated using a frontier-based exploration strategy. The agent’s next position is then determined by combining these candidate points with the exploration direction output by the Controller. With this global scene representation, we integrate various tools to acquire additional scene information and employ multi-step reasoning to fully utilize the collected observations. As a result, ToolEQA achieves superior performance.
>
> >**W1f:** The work invokes a VLM in every step for low-level navigation commands ("turn left"), compared to prior works (eg. GraphEQA) that predict navigation targets offloading navigation to low-level planner. The effect of this choice on results is also unclear.
>
> **R:** We are not offloading navigation to the low-level planner; instead, the high-level plan and low-level plan are integrated. At the beginning of each task, we first generate a plan (as shown below), which provides high-level guidance. We then use GoNextPoint to perform direction-level action selection.
> ```
> 1. Go to the office room or workspace area to locate the ergonomic office chair positioned at the end of the desk. Carefully check if someone is currently sitting in the chair to determine if it is occupied or unoccupied. If needed, approach the chair to observe any signs of occupancy such as a personal item left on it or wear marks.
> 2. Once the observation of the chair is complete and you have confirmed its occupancy status, you will have the necessary information to answer the question regarding whether the ergonomic office chair is currently occupied or unoccupied.
> ```
> We would like to clarify that comparing the advantages and disadvantages of different types of planners is not the focus of our work, but rather solving the problems of long-horizon reasoning and scene generalization in EQA. Our framework is designed as an integrated system, and the experiments demonstrate the superiority of ToolEQA as a whole.
>
>
> >**W1g:** While the work includes and additional test set with unseen scenes, it is not clear if the reported generalization (similar performance on seen and unseen scenes) is due to the unseen scenes having fewer target objects (Fig. 7c). It might be useful to break down the performance on the number of target objects to see if it has an effect on the performance of different splits (as hypothesized in the paper in L310).
>
> **R:** We would like to apologize for our oversight. In Fig.7c (updated to Fig.8a in the new version), we mistakenly labeled the number of thought steps as the number of objects. We have corrected this in the updated manuscript.
>
> We would like to clarify that the x-axis in Fig.7c (updated to Fig.8a in the new version) represents the number of thought steps, while the y-axis denotes the difference in the proportion of sample counts. This figure illustrates the difference in the distribution of thought-step counts between the seen and unseen splits.
> In addition, we provide supplementary statistics on the number of samples with different object counts in the seen and unseen splits (see Appendix Fig. 8b).
> As shown in Table.4, we evaluate tasks success rate with different numbers of targets to eliminate the influence of target quantity.
>
> **Table 4: Success rate with different numbers of targets.**
> |Number of objects|EQA-RT-Seen|EQA-RT-Unseen|
> |-|-|-|
> |1|60.7|60.93|
> |2|58.58|59.42|
> |3|54.83|57.67|
> |4|51.17|52.64|
> |5|42.72|49.84|
>
> The above results demonstrate that our method indeed has the ability to generalize to new scenarios, instead of relying on the object numbers.

---

> ### Author Response · Authors · 2025-11-24
>
> >**W2:** Multi-step reasoning with tool usage has been explored to great extent in non-embodied settings (ViperGPT, VisProg, T3-Agent) and multi-step reasoning has been explored for embodied question answering (Fine-EQA). As a result the technical contributions of the work are limited, i.e., applying T3-Agent to embodied QA.
>
> **R:** Our work is not a simple adaptation of existing methods; rather, we introduce substantive technical innovations in several key aspects:
> - Embodied tasks require exploration in physical environments, meaning that an embodied agent must possess autonomous navigation and spatial understanding capabilities. To support this, we maintain a voxel-based scene representation inside the tool, enabling the agent to reason about its own state and spatial relationships. Although we keep this representation within the tool for the sake of overall framework scalability, it is accessed at every reasoning step and can therefore be regarded as an integral component of the ToolEQA framework.
> - Due to the partially observable nature of embodied tasks, an embodied agent cannot obtain useful information at every reasoning step as non-embodied agents typically do. This leads to significantly longer reasoning trajectories (from 3–5 steps to 15-30 steps). To address the data annotation challenges posed by such long-horizon reasoning, we propose an automated data generation method.
>
> In summary, our work is far more than simply applying T3-Agent to EQA; instead, we provide new technical contributions tailored to the fundamental challenges of embodied reasoning.
>
>
> >**Q1:** The descriptions of tools are missing. For example, it is not clear how VisualQA or SegmentInstance are implemented and used.
>
> **A:** As shown in Table 5, we provide detailed descriptions of the implementation details of each tool.
>
> **Table 5：Details of tool implementation.**
> | Tool             | Implement |
> | - | - |
> | GoNextPoint      | We reconstruct the 3D scene into a voxel representation and project the voxels onto a 2D map. Based on this map, we employ a frontier-based exploration strategy to obtain multiple candidate exploration directions, and the final direction is selected according to the input parameters. |
> | ObjectLocation2D | GroundingDino [1] |
> | ObjectLocation3D | DetAny3D [2] |
> | ObjectCrop       | Python |
> | SegmentInstance  | SAM2 [3] |
> | VisualQA         | Qwen2.5VL [4]  |
> | FinalAnswer      | Qwen2.5VL [5] |
>
> During the initialization phase of ToolEQA, all tools are instantiated. The controller’s system prompt includes the full specification of each tool (method names, functional descriptions, inputs, and outputs). At runtime, the controller directly outputs Python code to invoke the tools.
>
> [1] Liu S, Zeng Z, Ren T, et al. Grounding dino: Marrying dino with grounded pre-training for open-set object detection[C]//European conference on computer vision. Cham: Springer Nature Switzerland, 2024: 38-55.
>
> [2] Zhang H, Jiang H, Yao Q, et al. Detect anything 3d in the wild[J]. arXiv preprint arXiv:2504.07958, 2025.
>
> [3] Ravi N, Gabeur V, Hu Y T, et al. Sam 2: Segment anything in images and videos[J]. arXiv preprint arXiv:2408.00714, 2024.
>
> [4] Bai S, Chen K, Liu X, et al. Qwen2. 5-vl technical report[J]. arXiv preprint arXiv:2502.13923, 2025.
>
>
> >**Q2:** It would be useful to include the prompts used for data generation and the VLM-based controller.
> >**Q6:** It would be useful to include prompts for the controller and the data generation pipeline.
>
> **A:** Thanks for the suggestions. We have updated the appendix (section A.6) and included all the prompts used for data generation and inference.
>
> >**Q3:** Have the authors considered using the Exploration-Answer Consistency (EAC) metric from FineEQA? Why do authors not use a consistent set of metrics cross all benchmarks?
>
> **A:** The EAC metric is designed to evaluate whether the single image used to answer a question truly contains the key information while considering the exploration distance. However, in the case of complex tasks (tasks that require the agent to explore multiple regions and collect multiple targets), EAC ignores whether the information for multiple targets involved in the question was actually obtained during the exploration process. Therefore, we introduce recall into our evaluation metric.
>
> Additionally, as of the time of submission, some methods have not been open-sourced. To ensure a fair comparison with these methods, we follow the evaluation metrics used in their respective works. We show the Fine-EQA results in Table 6, and we will update them in Table 5 of the manuscript.
>
> **Table 6: The result of Fine-EQA in different benchmarks.**
> | Model | HM-EQA (Succ.) | HM-EQA (L) | OpenEQA (Succ.)| OpenEQA (L) |
> | - | - | - | - | - |
> | Fine-EQA | 53.3 | 34.75 | 29.4 | 13.77 |

---

> ### Author Response · Authors · 2025-11-24
>
> >**Q5:** Can authors include more details and examples for the different categories of tasks (location-location, location-special)?
>
> **A:** We provide detailed explanations of the distinctions between each category tasks in the Table 7, along with a representative example. This context have been included in appendix table 9.
>
> **Table 7: Details of the distinctions between each task category.**
> | Type | Description | Example |
> | - | - | - |
> | Attribute-Size | Asking about the size of a single object or comparing the sizes of multiple objects. | Which object is taller, the rack against the wall next to the window or the lamp on the bedside table next to the bed? |
> |Attribute-Color|Asking about the color of a single object or comparing the colors of multiple objects.|Do the bottle and bag on the shelf next to the doll share similar color tones?|
> |Attribute-Special|Asking about other attributes of a single object (e.g., material, style).|What material is the countertop adjacent to the stove made of?|
> |Counting|Asking about the number of a certain object within a specified region.|What is the total number of pillows in the master and guest bedrooms combined?|
> |Distance|Asking about the distance from an object to the current position or the distance between two objects.|What is the distance between the heater in the bedroom and the lamp in the living room?|
> |Location-Location|Asking for the locations of a category of objects (possibly multiple locations).|In which room is the book currently placed?|
> |Location-Special|Asking for the location of a specifically described object (a particular, identifiable object).|In which corner of the kitchen is the large green plant inside a ceramic pot?|
> |Relationship|Asking about the relationship between two objects, including spatial relations and functional relations.|Are there any pillows on the sofa in the living room?|
> |Status|Asking about the state of an object.|Is the lamp in the bedroom turned on or off?|
>
>
> >**Q7:** For EQA task verification, why is it important to use both object detector and LLM? How do these compare?
>
> **A:** For the verification of EQA tasks, the object detector and the LLM are used for different aspects of evaluation. The object detector is employed to verify whether the objects mentioned in the question appear along the exploration path generated by A*. The LLM is used to assess four aspects:
> - Whether the question has a unique answer along the exploration path.
> - The degree of consistency between the options and the question.
> - Whether the options contain ambiguity or exhibit mutual confusion.
> - Whether the incorrect options are appropriate, based on the images along the exploration path.
>
> The object detector and the LLM are used for different aspects in the EQA task verification process, and using both of them is necessary.
>
> >**Q8:** How efficient is the data generation procedure? What fraction of trajectories get filtered out for each of the different reasons (L283-L286)?
>
> **A:** When generating the data using a single process, it takes approximately 8 days to complete the entire dataset.  About 38% of the generated data is filtered out, where rule-based validation (object detector) filters out about 13%, while LLM based validation filters out about 25%.

---

> ### Author Response · Authors · 2025-11-24
>
> >**Q9:** L320: It is not clear how finetuning to predict the final answer would hurt the model? Can the authors elaborate more on how this encourages the model to use its biases by ignoring the context?
>
> **A:** First, the supervision signal provided by the FinalAnswer is very coarse: it only constrains the final natural language answer, without guiding the model on how to arrive at it. EQA tasks involve complex environments with multiple objects, rooms, occlusions, and varying lighting, yet the FinalAnswer is often just a single word or short phrase. If the model is trained to generate the final answer directly from the input text, it can exploit statistical biases in the training distribution (e.g., certain objects typically appear in specific rooms or have common attributes) to guess the answer without truly processing the observed images or performing tool calls. This leads to shortcut learning, where the model relies on semantic priors and ignores the embodied context.
>
> Second, directly predicting the FinalAnswer does not effectively encourage reasoning. ToolEQA is designed for the model to explicitly generate a reasoning chain, including exploration, observation, tool invocation, and intermediate conclusions. If the final answer serves as the primary supervision signal, the model has no incentive to produce meaningful intermediate reasoning or execute the action–observation–reasoning loop, yet it can still achieve low training loss. Consequently, the model may skip tool calls, bypass environment exploration, and generate the final answer based solely on semantic biases.
>
> >**Q10:** The length of trajectories is lower than the trajectories -- an average of 12.69 (L321). Based on the distances in Table 6, does this mean the agent moves by more than 1m in each step? How does this not prevent from taking finer steps for moving closer to objects or navigating cluttered scenes?
>
> **A:** This does not mean that the agent moves more than 1m with each step. The values reported in Table 6 are averages. In practice, the agent’s step length varies depending on the environment’s spatial layout: in more cluttered scenes, each step tends to be shorter, whereas in more open areas, the agent may travel farther per step.
>
> >**Q11:** Some typos and questions in writing.
>
> **A:** We sincerely thank the reviewer for pointing out several typos. We will carefully address all of them in the next version of the paper.

---

### Official Review · Reviewer_8Z79 · 2025-11-02

**Soundness:** 2
**Presentation:** 3
**Contribution:** 2
**Rating:** 6
**Confidence:** 4

**Summary:**

This paper focuses on the embodied QA which requires a robot to navigate a home environment to find answer for the provided question. This paper focuses on an agentic setup to achieve this. This agentic setup is also the main contribution of the paper. Overall, the agent (robot) has a set of tools at it’s disposal. These tools include GoNextzPoint, ObjectLocation2D, etc. — some of these tools are used for navigation planning, while others are used for visual information extraction. Prior approaches often end up training models directly to solve the task end-to-end (often with VLMs). By contrast, the current paper aims to use an agentic setup to learn how to do the task using the provided tools. To achieve this, the paper focuses on creating a dataset with the appropriate tool calls. For this, the paper curates the appropriate questions given a scene, from these questions important objects are extracted and paths to reach close to these objects are planned using A*. Once this trajectory has been created, GPT-4O is used to annotate each step path with a reasoning/thought trace ($t_i$) and a code output $o_i$. This data is then used to FT a VLM (Qwen-7B). Experiments show that the finetuned models perform better than zero-shot models with access to the agent API.

**Strengths:**

The paper focuses on an important problem of using VLM agents to solve embodied QA. Overall, the paper is well written. The main agentic contribution also is novel and seems to provide some potential benefit over baselines.

**Weaknesses:**

Questions

4.3 details: I think the most important part of the paper is the dataset curation strategy. While there are sufficient details around how the questions and paths are generated, there is insufficient details around how the reasoning / thought traces were curated. For example, reasoning trace is also generated when the model/agent should output “Move Left” — how does the thought trace look for this kind of action? When generating the GT data do you provide all the history (with all the observations) to the model? The prompts used for GT data curation should be in the appendix.

Also, when we use the trajectory from A* we do have the GT for some actions (e.g. moving actions such as GoNextPoint), but how do you generate GT for the model that it should use other tools (e.g. ObjectCrop), why should GPT-4o use this tool? It is quite unclear how the model is choosing the right tool. This is an extremely important detail which is completely missing from the paper. How do you ensure that these tool calls are optimal? Also, some of these tool calls e.g. ObjectCrop can conflict with going closer to the target object (and zooming into the right object) so there is some weird design choice here.

*Ablation on training on thoughts*: Currently, the paper finetunes the model on the code output as well as the intermediate thoughts. What happens if we only finetune on the code (which is I assume the desired output) instead of the thought and maybe do some small RL to let the model curate it’s own thought. It would be interesting to see how the model performs then.

How do the tool calls look like for the zero-shot model (GPT-40 and QWEN-2.5 VL)

*Results:* There is very little performance difference (using e_path @5) metric between GPT-4o and the FT-VLM for MCQ questions. Are most of the GPT failures due to navigation actions being incorrect and the agent not exploring enough or due to some other scenarios. I am wondering if the agent has an access to a navigation tool which can plan for longer and the agent doesn’t have to output every point to go to next, how would the performance change.

**Questions:**

see above

---

> ### Author Response · Authors · 2025-11-24
>
> >**Q1:** The most important part of the paper is the dataset curation strategy. While there are sufficient details around how the questions and paths are generated, there is insufficient details around how the reasoning / thought traces were curated. For example, reasoning trace is also generated when the model/agent should output “Move Left” — how does the thought trace look for this kind of action? When generating the GT data do you provide all the history (with all the observations) to the model? The prompts used for GT data curation should be in the appendix.
>
> **A:** For non-key steps (i.e., steps that do not involve object information essential for answering the question), we use the A* algorithm to determine the next step in the GT exploration path. This exploration direction, along with the current RGB image, is then fed into the VLM, which generates the thought process in reverse.
>
> For key steps, we select a subset of tools based on the task category, then input both the selected tool subset and the critical information required at that step into the model, prompting it to reverse-engineer the thought process. This reverse reasoning approach has been shown to be effective in previous studies [1,2].
>
> In generating GT data, we input all historical information and current observation as context to LLM to obtain thought and code.
> We have added all prompts to the Appendix A.6.
>
> [1] Xue T, Wang Z, et al. Rcot: Detecting and rectifying factual inconsistency in reasoning by reversing chain-of-thought[J]. arXiv preprint arXiv:2305.11499, 2023.
>
> [2] Chen J, Wang Z, et al. Reverse thinking makes llms stronger reasoners[C]//Proceedings of the 2025 Conference of NAACL: Human Language Technologies. 2025: 8611-8630.
>
> >**Q2:** when we use the trajectory from A* we do have the GT for some actions (e.g. moving actions such as GoNextPoint), but how do you generate GT for the model that it should use other tools (e.g. ObjectCrop), why should GPT-4o use this tool? It is quite unclear how the model is choosing the right tool. This is an extremely important detail which is completely missing from the paper. How do you ensure that these tool calls are optimal? Also, some of these tool calls e.g. ObjectCrop can conflict with going closer to the target object (and zooming into the right object) so there is some weird design choice here.
>
> **A:** For each key step, we provide a task-specific tool subset, predefined based on the question type (we updated predefined tool subset in Appendix Table. 9).
> We prompt GPT-4o with the information it needs to acquire at that step, enabling it to autonomously select the appropriate tools from tool subset.
> In addition, the use of the `ObjectCrop` tool is not in conflict with approaching the target object. Even when the agent is very close to the target, it will inevitably observe surrounding irrelevant objects, and such redundant information may negatively affect reasoning performance [1]. Therefore, we use the `ObjectCrop` tool to filter out these distractions.
> At the final key step, the agent must also call the `finalanswer` tool to produce the final answer.
>
> [1] Zhang J, Khayatkhoei M, Chhikara P, et al. Mllms know where to look: Training-free perception of small visual details with multimodal llms[J]. arXiv preprint arXiv:2502.17422, 2025.
>
> >**Q3:** The paper finetunes the model on the code output as well as the intermediate thoughts. What happens if we only finetune on the code (which is I assume the desired output) instead of the thought and maybe do some small RL to let the model curate it’s own thought. It would be interesting to see how the model performs then.
>
> **A:** Thanks for your suggestions. We conduct a small-scale exploration of reinforcement learning methods. Specifically, we design two types of rewards for the agent: (1) a navigation reward, which provides positive feedback when the agent moves closer to the target object and negative feedback when it moves away or collides; (2) an answering reward, which grants a high reward for producing the correct final answer. The action space includes basic navigation actions (e.g., moving forward, turning) as well as tool-use actions (e.g., objectlocation3d). The agent is trained with the PPO algorithm, interacting continuously with the environment and updating its policy network based on reward signals.
>
> However, this RL method did not obtain satisfactory results. We consider this is due to two main factors: (1) reinforcement learning generally requires substantially greater computational resources, but the rebuttal period was too rushed; (2) for challenging EQA tasks, it is difficult to sample the correct trajectory, resulting in low efficiency of RL training; (3) RL method demands careful algorithmic design, such as crafting an appropriate reward function.
>
> Even so, we believe that reinforcement learning remains a promising training paradigm, and we plan to further investigate RL-based approaches in future work.

---

> ### Author Response · Authors · 2025-11-24
>
> >**Q4:** How do the tool calls look like for the zero-shot model (GPT-4o and QWEN-2.5 VL)
>
> **A:** We present the tool calls made by different models in the multiple steps for the same question.
>
> Question: Which object is taller, the chair at the corner of the dining area next to the marble table or the kitchen cabinet adjacent to the stainless steel refrigerator and facing the dining table?
> GPT-4o:
> ```python
> # Step 2:
> position, size = ObjectLocation3D(object='kitchen cabinet', image_path='next_point_2.jpg')
> print(f'The information of kitchen cabinet is: position is {position},  size (Length, width, height) is {size}.')
>
> # Step 4:
> position, size = ObjectLocation3D(object='chair', image_path='next_point_4.jpg')
> print(f'The information of chair is: position is {position},  size (Length, width, height) is {size}.')
>
> # Step 5:
> chair_height = 0.93
> cabinet_height = 0.44
> if chair_height > cabinet_height:
>     answer = "A. The chair is taller"
> elif cabinet_height > chair_height:
>     answer = "B. The kitchen cabinet is taller"
> else:
>     answer = "C. They are the same height"
> print(answer)
> ```
> QWEN-2.5 VL:
> ```python
> # Step 2:
> position, size = ObjectLocation3D(object='kitchen cabinet', image_path='next_point_2.jpg')
> print(f'The information of kitchen cabinet is: position is {position},  size (Length, width, height) is {size}.')
>
> # Step 5:
> position, size = ObjectLocation3D(object='chair', image_path='next_point_5.jpg')
> print(f'The information of chair is: position is {position},  size (Length, width, height) is {size}.')
>
> # Step 6:
> print('The height of the chair is 0.94, and the height of the kitchen cabinet is 0.5. Since 0.5 < 0.94, the chair is taller.')
> ```
> It is evident that the first two steps produce similar tool calls in the two models, but in the final step GPT-4o generates a comparatively more reasonable tool invocation. Thus, the experimental results in Table 1 show that GPT-4o substantially outperforms Qwen2.5-VL. We attribute this to differences in reasoning ability, which lead to different decisions when determining the exploration direction, thereby affecting both exploration efficiency and success rate.
>
> >**Q5:** There is very little performance difference (using e_path @5) metric between GPT-4o and the FT-VLM for MCQ questions. Are most of the GPT failures due to navigation actions being incorrect and the agent not exploring enough or due to some other scenarios. I am wondering if the agent has an access to a navigation tool which can plan for longer and the agent doesn’t have to output every point to go to next, how would the performance change.
>
> **A:** In the multiple-choice setting, the answer options provide strong priors, allowing the agent to infer the correct answer from the textual cues even when there are imperfections in visual understanding or navigation. This reduces the impact of the underlying large model on overall performance.
>
> Furthermore, an analysis of GPT’s failure cases shows that most failures are due to insufficient exploration: successful episodes have an average of 18.6 exploration steps, whereas failed episodes average only 9.6 steps, resulting in the agent missing critical information needed to answer the questions.
>
> Finally, in the current version of ToolEQA, the planner generates a long-term navigation goal (as shown below), and the controller determines the next exploration actions based on this goal. This means that our navigation tool is not limited to purely local navigation. We find your suggestion of directly using a navigation tool capable of planning longer trajectories without specifying every intermediate step very interesting, and we plan to explore this approach in future work.
>
> ```
> 1. Go to the office room or workspace area to locate the ergonomic office chair positioned at the end of the desk. Carefully check if someone is currently sitting in the chair to determine if it is occupied or unoccupied. If needed, approach the chair to observe any signs of occupancy such as a personal item left on it or wear marks.
> 2. Once the observation of the chair is complete and you have confirmed its occupancy status, you will have the necessary information to answer the question regarding whether the ergonomic office chair is currently occupied or unoccupied.
> ```

---

### Official Review · Reviewer_WRua · 2025-11-05

**Soundness:** 3
**Presentation:** 3
**Contribution:** 2
**Rating:** 4
**Confidence:** 4

**Summary:**

This paper proposes ToolEQA, a framework for Embodied Question Answering (EQA) where an agent uses external tools to navigate and gather information within 3D environments. By breaking down tasks into explicit reasoning steps and generating code to call these tools, the agent achieves more efficient exploration. To train their model, the authors created EQA-RT, a large-scale dataset with automatically generated reasoning trajectories. Experiments show ToolEQA achieves strong performance, particularly in reducing the exploration path length required to answer questions.

**Strengths:**

1.The ToolEQA framework uses explicit reasoning to guide its actions, resulting in shorter navigation paths and more efficient task completion compared to previous methods.
2.The paper contributes EQA-RT, a large-scale dataset with detailed reasoning trajectories. It ingeniously integrates 3D object detection, GPT-4o for question generation, and A* for optimal path planning, all validated by a multi-level verifier, to produce high-quality training data. The automated pipeline used to generate this data is a valuable resource for the community, enabling the creation of complex, structured EQA tasks.

**Weaknesses:**

1.The core ToolEQA framework is conceptually indistinct from established tool-augmented LLM paradigms like ReAct and Toolformer. Its "thought-code-observation" loop is a direct application of this existing work, making the contribution feel more like a conceptual repackaging for the EQA domain rather than a fundamental innovation. The authors should more clearly articulate what unique, non-trivial architectural adaptations were made to the framework specifically for the challenges of embodiment, beyond simply creating a new set of tools.
2.The paper's most significant and original contribution appears to be the EQA-RT dataset and its automated generation pipeline. While this is an impressive engineering system, it does not in itself constitute a novel methodology. The work's core value lies in the resource it provides to the community, which raises questions about its fit as a methods paper, as the novelty in the learning algorithm or agent architecture is minimal.

**Questions:**

1.The ToolEQA framework closely mirrors existing tool-augmented LLM paradigms. Could you elaborate on the unique challenges that embodiment introduces and how ToolEQA's architecture was specifically adapted to address these? For instance, how does the system handle or recover from incorrect predictions from its perception tools? What is the core innovation beyond applying a known method to a new domain?
2.A significant portion of the paper's contribution is the EQA-RT dataset, which is generated by a pipeline that naturally creates tasks solvable by a tool-based agent. How can we be confident that the agent is learning a generalizable reasoning skill rather than simply overfitting to the specific patterns of the EQA-RT generation process? Have you considered evaluating its performance on tasks that are explicitly designed to challenge the limits of its predefined toolset?
3.The current work relies on a small, fixed set of tools. How do you envision the framework scaling to more open-ended real-world scenarios? How does the agent react when faced with a task where its tools are insufficient? Can the fine-tuned controller generalize to use new tools effectively without requiring SFT? Does the framework support the autonomous discovery or learning of new tool functionalities?
4.Regarding the reasoning process itself, is a higher frequency of tool calls always beneficial? Have you analyzed the potential for redundant tool calls, where the agent re-requests information it already possesses or could have inferred? Could you provide an analysis of the correlation between the number of tool invocations per task and the final success rate? This could help clarify whether the agent's reasoning is efficient or simply exhaustive.

---

> ### Author Response · Authors · 2025-11-24
>
> >**W1:** The core ToolEQA framework is conceptually indistinct from established tool-augmented LLM paradigms like ReAct and Toolformer. Its "thought-code-observation" loop is a direct application of this existing work, making the contribution feel more like a conceptual repackaging for the EQA domain rather than a fundamental innovation. The authors should articulate what unique, non-trivial architectural adaptations were made to the framework for the challenges of embodiment.
>
> >**Q1.** Could you elaborate on the unique challenges that embodiment introduces and how ToolEQA's architecture was specifically adapted to address these? For instance, how does the system handle or recover from incorrect predictions from its perception tools? What is the core innovation beyond applying a known method to a new domain?
>
> **R:** ToolEQA is not a simple application of ReAct or other tool-augmented LLM approaches to the EQA task. Instead, it draws on the core ideas of ReAct to specifically address the unique challenges of long-horizon reasoning and scene generalization in EQA. Concretely, transferring the ReAct paradigm to embodied settings needs to solve the following challenges:
> 1. Embodied agents need to develop spatial perception capabilities in the real physical world, inferring 3D structures from first-person local observations, understanding object relationships, occlusions, and accessibility, while maintaining consistent spatial cognition during movement. In contrast, non-embodied agents do not involve observation and long-horizon reasoning of physical space.
> 2.  Each observation of embodied agent provides only a limited field of view, and exploration often yields irrelevant information. As a result, the reasoning chain becomes substantially longer (typically 15–30 steps, compared to 3–5 steps in conventional tool-augmented LLM tasks), leading to a significant increase in reasoning and training difficulty and requiring more high quality data.
>
> These factors fundamentally differentiate embodied agents from traditional tool-augmented LLM methods. To address these challenges, ToolEQA introduces the following design.
>
> To enhance the controller’s spatial awareness and reasoning capabilities, we maintain an internal voxel-based scene representation. With the voxel-based scene representation, we can use external tools to abstract its spatial information into high-level structured descriptions for the LLM. For example, using ObjectLocation3D to supply objects’ 3D positions and dimensions, and GoNextPoint to indicate navigable directions. This mechanism tightly integrates the physical environment with the LLM-based controller, enabling more accurate spatial understanding and reasoning.
>
> In addition, because increasing the number of exploration steps significantly raises training difficulty, we aim to address this challenge by scaling up the dataset and improving data quality. To this end, we propose an automated data generation pipeline: we first sample object information in the scene (such as size and coordinates), then use GPT-4o to automatically generate question–answer pairs; next, we compute the shortest path using the A* algorithm considering the challenging trajectory length, and use GPT-4o to generate thoughts augmented with tool calls; finally, a dedicated task and trajectory validator filters out low-quality or incorrect samples. This automated data generation process effectively meets the need for large-scale, high-quality data for embodied agents.
>
> > **W2:** The paper's most significant and original contribution appears to be the EQA-RT dataset and its automated generation pipeline. While this is an impressive engineering system, it does not in itself constitute a novel methodology. The work's core value lies in the resource it provides to the community, which raises questions about its fit as a methods paper, as the novelty in the learning algorithm or agent architecture is minimal.
>
> **R:** We believe ToolEQA makes significant and original contributions.
> First, we innovatively apply the ReAct paradigm to the EQA task, successfully addressing challenges in interpretability and scene generalization in physical environments.
> Second, to address challenges in spatial understanding and automatic navigation, we maintain an internal voxel-based scene representation and use external tools to provide structured spatial descriptions, enabling the LLM to perceive object positions, dimensions, and navigable directions. By scaling up the dataset, generating multi-step reasoning augmented with tool calls, and validating trajectories, we enable the model to perform long-horizon reasoning and solve complex real-world tasks.
> Finally, our automated data generation method not only reduces the cost of data collection but also provides a scalable and reusable solution for embodied tasks that require long-horizon reasoning. Based on these innovation in methodology, we argue that this work can be regarded as a method paper.

---

> ### Author Response · Authors · 2025-11-24
>
> >**Q2:** A significant portion of the paper's contribution is the EQA-RT dataset, which is generated by a pipeline that naturally creates tasks solvable by a tool-based agent. How can we be confident that the agent is learning a generalizable reasoning skill rather than simply overfitting to the specific patterns of the EQA-RT generation process? Have you considered evaluating its performance on tasks that are explicitly designed to challenge the limits of its predefined toolset?
>
> **A:** ToolEQA does not simply overfit to a fixed reasoning pattern; rather, it genuinely acquires reasoning skills. As shown in Figure 10 of Appendix section A.2, we analyzed the sequence distribution of tool usage on EQA-RT-Seen and EQA-RT-Unseen, and observed significant differences between them. This indicates that ToolEQA selects and invokes tools based on the current observation, rather than following a predetermined reasoning sequence.
>
> In this paper, we propose the first tool-enhanced EQA framework, while we do not consider the case where solving tasks requires new tools beyond the predefined tool set. Thank you for raising such an important point. During the rebuttal period, we observe that our method can generalize to new tools that were not seen during training, but the definition of new tools still requires human intervention.（Details can be found in our response to Q3） In the future work, we plan to break down the limitations of requiring manually defined tools by designing an agent that could autonomously acquire new tools from the internet or through interaction with humans. We have added the analysis to the Limitations section.
>
> >**Q3:** The current work relies on a small, fixed set of tools. How do you envision the framework scaling to more open-ended real-world scenarios? How does the agent react when faced with a task where its tools are insufficient? Can the fine-tuned controller generalize to use new tools effectively without requiring SFT? Does the framework support the autonomous discovery or learning of new tool functionalities?
>
> **A:** Given a predefined tool set, ToolEQA is capable of zero-shot tool usage. When faced with more open real-world scenarios with new tools (untrained but given in predefined toolset), it can adapt to these tasks simply by calling new tools appropriately. To verify it, we manually designed a tasks that can only be completed using new provided tools outside the trained tool set. As shown in Figure 11 of Appendix section A.2, ToolEQA is still able to invoke these untrained tools. This demonstrates that our fine-tuned model can effectively generalize to new tools without requiring additional supervised fine-tuning.
>
> Considering the source of the new tools, ToolEQA currently does not possess the capability of autonomous discovery or learning new tool functionalities, and it relies on human definition. In the future work, we aim to design an agent that could autonomously acquire new tools from the internet or through interaction with humans. We have added the analysis to the Limitations section.
>
>
> >**Q4:** Regarding the reasoning process itself, is a higher frequency of tool calls always beneficial? Have you analyzed the potential for redundant tool calls, where the agent re-requests information it already possesses or could have inferred? Could you provide an analysis of the correlation between the number of tool invocations per task and the final success rate? This could help clarify whether the agent's reasoning is efficient or simply exhaustive.
>
> **A:** Frequent tool usage is not always beneficial. As shown in Figure.9 of Appendix section A.2, the success rate initially increases with the number of tool calls but then decreases, indicating that redundant tool usage does occur. Therefore, the frequency of tool invocation should be maintained within a reasonable range.

---

### Author Response · Authors · 2025-12-02
**Summary of Author Rebuttal [Part II]**

## Secondary Concerns:
- **Missing details about data generation and controller prompts (Reviewer 8Z79, zT4c):** We have included all prompts for data generation and inference in Appendix A.6, providing complete transparency for reproducibility.
- **Data generation pipeline efficiency and filtering statistics (Reviewer zT4c):** The pipeline takes ~8 days for full dataset generation, with 38% filtered out (13% by object detector, 25% by LLM validation).
- **Different roles of object detector and LLM in task verification (Reviewer zT4c):** We clarified that object detector verifies object presence along paths, while LLM assesses answer uniqueness, option consistency, ambiguity, and appropriateness.
- **Figure labeling errors and impact of target object counts (Reviewer zT4c):** We corrected Fig. 7c (now Fig. 8a) and added analysis showing consistent performance across different target counts (60.7% vs 60.93% for 1 object, 42.72% vs 49.84% for 5 objects across seen/unseen splits).
- **Tool definition details and scope of code block functionality (Reviewer zT4c, mden):** We provided complete tool implementation details (Table 8) and examples showing code blocks perform logical operations beyond print, such as size comparisons and conditional reasoning.

## Summary of Revisions:
We have updated the revised version with all additional discussions and experiments conducted during the rebuttal. Specifically, we will:
- Include all prompts for data generation and inference in Appendix A.6.
- Add detailed tool implementation descriptions and examples in Appendix A.4.
- Incorporate ablation results on thought, tool, and planner components in Section 5.2.
- Update efficiency metrics and zero-shot vs. fine-tuned comparisons across all benchmarks in Tables 3-5.
- Correct all typos and improve clarity throughout the manuscript.

Thank you again for your valuable contributions to strengthening this work.

Best regards,

Authors

---

### Author Response · Authors · 2025-12-02
**Summary of Author Rebuttal [Part I]**

Dear all:

We sincerely thank you for your detailed reviews, valuable feedback, and thoughtful discussions throughout the review process. We are truly grateful for your recognition and constructive comments, which have helped us significantly improve the paper. We especially appreciate the AC's support in facilitating productive discussions.

## Summary of Recognized Strengths:
We are grateful for the reviewers' recognition of our work and its key contributions:
1. Well-structured paper with clear motivation and method description. (Reviewer 8Z79, zT4c, mden)
2. Novel integration of tool-augmented reasoning into embodied question answering. (all reviewers)
3. Introduction of the EQA-RT dataset and automated generation pipeline. (Reviewer WRua, mden)
4. Promising performance on multiple benchmarks. (all reviewers)
5. Significant improvements in success rate and exploration efficiency. (all reviewers)

## Primary Concerns:
**1. Novelty (Reviewer WRua, zT4c)**
We thank the reviewers for acknowledging that our automated data generation pipeline and the EQA-RT dataset represent the significant contribution of this work. Generating high-quality, long-horizon reasoning trajectories for embodied settings is particularly challenging due to partial observability, the need for spatial consistency, and the scalability of annotation. Our pipeline, which integrates 3D detection, GPT-4o for question generation, A* for path planning, and multi-stage verification, provides a systematic and reproducible solution to this problem. It enables the community to generate complex EQA tasks at scale.

Beyond the dataset, the ToolEQA framework itself introduces key adaptations tailored for embodied reasoning. Unlike conventional tool-augmented LLMs that operate in fully observable settings with short reasoning chains, ToolEQA is designed to handle long-horizon reasoning under partial observability. It maintains an internal voxel-based scene representation for spatial grounding and dynamically selects tools based on real-time observations rather than following fixed scripts. These embodied-specific innovations distinguish ToolEQA from a straightforward application of existing tool-augmentation paradigms.

**2. Generalization (Reviewer WRua, 8Z79, zT4c)**
To ensure that ToolEQA learns generalizable reasoning skills rather than overfitting to dataset-specific patterns, we analyzed the distribution of tool-usage sequences across EQA-RT-Seen and EQA-RT-Unseen. The results (Figure 10) show significantly different tool-call patterns between the two splits, indicating that the agent selects tools based on current observations and task context, not merely memorizing training sequences. Additionally, we evaluated performance across tasks with varying numbers of target objects (1 to 5) and observed consistent success rates between seen and unseen scenes.
In addition, we designed an experiment to test generalization to new, unseen tools. Generalization to new tools was tested by introducing unseen tools during inference; ToolEQA successfully invoked them without additional fine-tuning, demonstrating flexible tool-use capability.

**3. Comprehensive Analyses (All reviewers)**
- We added more ablation studies on several components of ToolEQA. Ablation studies show that removing thoughts reduces success rate from 56.1% to 48.6%, removing tools reduces it to 41.9%, and removing the planner increases path length from 16.63 to 25.97, confirming that each component is essential.
- We performed a quantitative analysis of tool-call efficiency. Tool-call efficiency analysis (Figure 9) reveals that success rate initially rises with tool usage but declines beyond 10–15 calls, indicating that our agent learns to avoid redundant invocations.
- We evaluated exploration efficiency across multiple established benchmarks. Efficiency metrics across benchmarks show that ToolEQA achieves substantially shorter exploration distances (e.g., 6.96m on OpenEQA vs. 11.32–25.45m for baselines) while maintaining competitive success rates.
- These analyses were integrated to form a comprehensive validation. These analyses collectively confirm that ToolEQA performs efficient, generalizable, and component-aware reasoning in embodied settings.

---

### Meta-Review · Area_Chair_R71d · 2026-01-05

**Summary:**

This work proposes to combine multi-step reasoning with VLMs and tool use for embodied question answering, as well as a dataset generation pipeline for training such a system. Reviews had high variance: 2, 4, 6, 8. The concerns about the paper fell into two categories: low novelty due to combining existing ideas in a fairly straightforward manner for the EQA problem (namely, reasoning with VLMs and equipping VLMs with tools), and concerns about insufficient ablations and experimental comparisons (mostly brought up by reviewer zT4c). The most positive review (score: 8) was quite short and vague, and did not give much information about the paper strengths (simply saying that the system was useful, results were good, etc.), therefore, I am discounting this review to some degree. The author rebuttal addressed some of the concerns by reviewer zT4c, who I would expect to raise their score modestly. I do not think the author rebuttal addressed the broader questions around the paper’s novelty, generality, and impact. The paper does an effective job at applying a well tested recipe (VLM reasoning + tool use) to the problem of embodied question answering, but doesn't seem to provide new insights beyond confirming that this recipe is reasonably effective on yet another task, subject to appropriately designed tools. That being said, the reviewers generally agreed that the proposed dataset could be useful to the community. This is a borderline paper for which I am recommending reject, but would also not mind if the paper is accepted.

**Reviewer Concerns:**

The questions about more fine-grained ablations and experimental details that were previously missing have partially been addressed. The concerns about novelty and broader impact have not been addressed in my opinion.

**Reviewer Scores:**

WRua: 4 -> 4

8Z79: 6 -> 6

zT4c: 2 -> 4

mden: 8 -> 8

---

### Decision · Program_Chairs · 2026-01-26

Reject